# PGS: Effective LLM Code Refinement via Property-Oriented and Structurally Minimal Feedback

**Lehan He** [* 1 2]  **Zeren Chen** [* 1 3]  **Zhe Zhang** [1]  **Xiang Gao** [1]  **Lu Sheng** [1]

## Abstract

Large Language Models (LLMs) excel at code generation, yet ensuring the functional correctness of their outputs remains a persistent challenge. Recent studies have applied Test-Driven Development (TDD) to refine code, leveraging execution feedback to guide the model toward correct solutions. However, such feedback is often noisy and uninformative, stemming from the scarcity of high-quality test cases and the abundance of noisy, auto-generated ones. In this work, we shift the focus from test-case generation to feedback quality. We introduce the **Property-Generated Solver** (**PGS**), a novel feedback-centric framework designed to generate highly effective feedback via two principles: it must provide semantic guidance beyond simple I/O mismatches through **property validation**, and be **structurally minimal**, to reduce cognitive load and isolate root causes. PGS operates by checking high-level program properties (*e.g.*, a sorting function must produce a non-decreasing sequence) then providing the simplest failing counterexample to the LLM. This property-driven, minimal feedback steers LLMs toward correct and generalizable solutions. Across diverse benchmarks, PGS demonstrates superior performance, achieving a bug fix rate 1.4x-1.6x higher than the strongest debugging-based approaches and establishing a new state-of-the-art in automated code refinement.

## 1. Introduction

Recent advances in LLMs have revolutionized automated code generation, enabling tools like GitHub Copilot to assist developers in translating natural language requirements into functional code (OpenAI, 2023; Bai et al., 2023; Zhu et al., 2024). However, ensuring the functional correctness of the generated code remains a critical challenge, representing a primary bottleneck to the reliable deployment of these models in real-world scenarios (Liu et al., 2024a). To bridge this gap, many have turned to the Test-Driven Development (TDD) (Jiang et al., 2023; Zhong et al., 2024; Shinn et al., 2024) framework for iterative refinement. TDD framework leverages a cycle of test execution and code modification, where outcomes like pass/fail status or error messages, serve as feedback to guide the LLM. This feedback-driven loop allows the model to progressively debug and enhance its initial code generation, steering it toward a correct solution.

However, the practical effectiveness of this feedback-driven loop is constrained by two fundamental problems. The first is the scarcity of high-quality test cases. While a straightforward remedy might be to use LLMs to generate more tests (Chen et al., 2023a; Liu et al., 2024b), this approach often leads to a "cycle of self-deception," where the test generator shares the same logical biases as the code generator. Furthermore, generating the correct output for a given test input (*a.k.a* test oracle) can be as difficult for an LLM as solving the original problem itself (Barr et al., 2014; Jain et al., 2025). Second, and more critically, the quality of the feedback itself has been largely overlooked. The mass generation of flawed tests, as mentioned, directly degrades feedback quality by providing an uninformative and noisy signal to the LLM. Moreover, even a test case that correctly identifies a bug may be detrimental if its feedback is too complex. As shown in Figure 1 (b), a lengthy, convoluted failing input generates an intricate execution trace. This creates a high cognitive load that can overwhelm the LLM's reasoning process and misguide its refinement attempts, leading to repair failure.

This raises a crucial research question: *How to construct high-quality, actionable feedback to enable robust code refinement for LLMs?* We argue the most effective feedback is not found by generating more tests, but by improving the intrinsic quality of feedback itself. An ideal feedback must be both **property-oriented** and **structurally minimal**, shifting the focus from finding bugs incidentally through mass test generation to a more deliberate process of semantic

*Equal contribution [1]School of Software, Beihang University, Beijing, China [2]Shanghai Innovation Institute, Shanghai, China [3]Shanghai AI Laboratory, Shanghai, China. Correspondence to: Lu Sheng <lsheng@buaa.edu.cn>.

*Proceedings of the 43rd International Conference on Machine Learning*, Seoul, South Korea. PMLR 306, 2026. Copyright 2026 by the author(s).

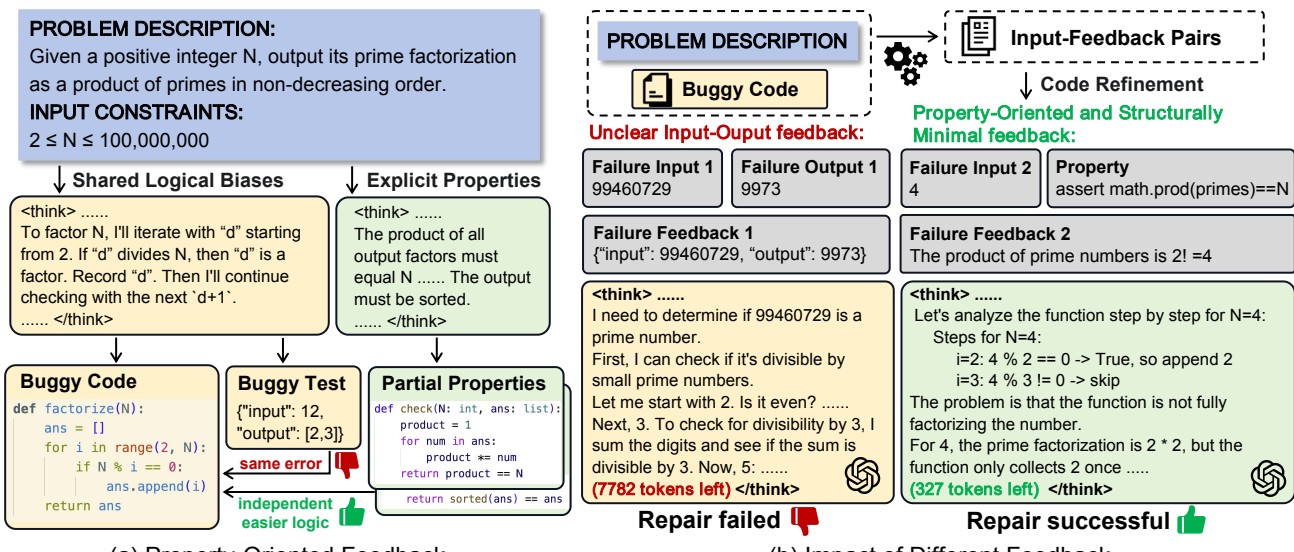

*Figure 1.* **The Principles of Effective Feedback for LLM Code Refinement.** **(a)** Property-driven validation avoids the pitfall of shared logical biases. While a buggy code generator often produces equally flawed I/O tests, property checks rely on simpler, independent logic, providing a more reliable signal for correctness. **(b)** A comparison of feedback impact. A complex, I/O-based counterexample creates high cognitive load and leads to repair failure. In contrast, our **property-oriented** and **structurally minimal** feedback uses the simplest failing input to provide a clear, actionable signal (*e.g.*, "The product of prime numbers is 2 != 4"), enabling a successful repair.

validation, as shown in Figure 1 (a). Specifically, we define these two principles as follows: **(1) Property-Oriented.** The feedback should move beyond concrete input-output pairs to validate fundamental program properties (Claessen & Hughes, 2000), *i.e.*, high-level, universal characteristics the code must satisfy for all valid inputs. For instance, a key property of any sorting function is that its output must be a non-decreasing sequence. This fundamentally addresses the "cycle of self-deception" problem due to *asymmetry of verification* (Wei, 2025), *i.e.*, verifying a solution is logically simpler than generating that solution. This ensures a reliable oracle even when generation fails, providing clear semantic guidance that helps the LLM to repair the root cause of errors. **(2) Structurally Minimal.** Rather than using a raw, complex failing test case, the feedback should be derived from the simplest possible counterexample that violates a property. This minimality isolates the error's root cause, removes distracting noise from the execution trace, and provides a clean, actionable signal tailored to an LLM's step-by-step reasoning. This directly addresses the cognitive load issue, preventing the model from being overwhelmed by convoluted failure scenarios (Liu et al., 2024c).

To operationalize these principles, we introduce the Property-Generated Solver (PGS), a multi-agent framework for feedback-centric code refinement. PGS employs a collaborative loop between a **Generator** agent, which produces and refines code, and a **Tester** agent, which crafts the property-oriented and structurally minimal feedback. Specifically, the Tester validates the generated code against high-level properties derived from the problem description. Upon

detecting a violation, it identifies the structurally minimal counterexample and formulates it into a concise, actionable signal for the Generator. This loop of targeted feedback and iterative refinement steers the solution toward functional correctness and generalizability. Across a diverse suite of benchmarks, from function-level synthesis (Chen et al., 2021; Austin et al., 2021) to complex competition-level problems (Jain et al., 2025; Li et al., 2022) and repository-level tasks (Jimenez et al., 2024), PGS consistently establishes a new state-of-the-art, demonstrating superior performance by achieving a bug fix rate **1.4x-1.6x higher** than even the most sophisticated debugging frameworks.

Our main contributions can be summarized as below:

- **A New Principle for Feedback Design.** We establish that effective feedback for LLM-based code refinement must be both property-oriented and structurally minimal. This shifts the focus from test quantity to feedback quality, addressing a critical dimension overlooked by existing TDD methods that often produce noisy and convoluted signals.

- **A Novel Feedback-Centric Multi-Agent Framework.** We design and implement the Property-Generated Solver (PGS), a multi-agent framework that operationalizes the proposed principles. PGS utilizes a collaborative loop between a Generator and a Tester agent to systematically construct high-quality feedback and guide the iterative refinement process.

- **State-of-the-Art Empirical Results.** We demonstrate through extensive experiments on multiple challenging

benchmarks that PGS significantly outperforms existing TDD-based methods. Our framework sets a new state-of-the-art for automated code refinement, showcasing the benefits of high-quality feedback.

## 2. Related Work

**LLM-Driven Code Refinement.** A significant research stream improves LLM-generated code through iterative refinement. Inspired by software development workflows (Jin et al., 2024; Xia & Zhang, 2023), these approaches guide LLMs using feedback from program execution. Existing methods differ mainly in how they obtain feedback, from lightweight execution errors or verbal reflections (Zhang et al., 2023; Shinn et al., 2024), to generated tests for candidate ranking (Chen et al., 2023a), and to structured debugging signals such as runtime states or multi-level error localization (Chen et al., 2023b; Zhong et al., 2024; Shi et al., 2024). Recent analyses further show that self-repair is not a guaranteed win: its gains can be modest once repair cost is counted, and its effectiveness is strongly bottlenecked by feedback quality (Olausson et al., 2024). Most existing refinement loops still rely on raw I/O mismatches, execution traces, or verbose debugging logs, which can identify failure but often provide limited semantic guidance. These findings motivate our focus on constructing more actionable feedback rather than merely adding another refinement loop. PGS differs from prior refinement systems by using property violations, rather than raw execution traces or I/O mismatches, as the primary repair signal.

**Specification-Based Test and Property Generation.** To overcome test scarcity, one line of research uses LLMs to generate I/O test cases (Chen et al., 2023a; Liu et al., 2024b). However, such methods face two practical challenges. First, producing the expected output for a generated input is an instance of the test oracle problem (Barr et al., 2014; Jain et al., 2025). Second, when the same or similar LLMs generate and assess candidate outputs, the assessment may reflect model-specific biases rather than independent evidence (Pan et al., 2024; Liu et al., 2024d). Property-based testing (PBT), popularized by QuickCheck (Claessen & Hughes, 2000), checks programs against general properties and reports counterexamples. This idea has a long history in software testing and verification, and recent work has explored LLMs for generating formal postconditions (Endres et al., 2024), property-based tests (Vikram et al., 2023), trustworthy synthesis checks (Key et al., 2022), property-retrieval-augmented unit tests (Zhang et al., 2024b), and class invariants (Sun et al., 2025). While property-based testing and counterexample minimization are established ideas in software engineering, their role in LLM-based code refinement remains underexplored. In our PGS framework, synthesized inputs do not require expected outputs; feed-

back is obtained by executing property checks rather than predicting test oracles. PGS uses LLM-generated properties within an iterative repair process: it extracts properties from natural language requirements, filters them against visible tests, probes candidate programs with synthesized inputs, and returns minimal property violations to guide code revision.

**Feedback Minimization for Program Refinement.** The principle of minimizing failure-inducing inputs, established by delta-debugging (Misherghi & Su, 2006), is crucial for automated refinement. In classical debugging, minimization improves human interpretability; in LLM refinement, it also controls how much irrelevant context enters the repair prompt. Recent work (Yang et al., 2025b) explores this for LLMs to reduce their cognitive load and mitigate issues like the "lost-in-the-middle" (Liu et al., 2024c) problem. Our work extends this idea to property-driven refinement: instead of minimizing inputs for I/O mismatches, PGS minimizes counterexamples violating high-level semantic properties. This setting does not require reference outputs for the synthesized inputs and raises a key question: what does "minimal" truly mean for an LLM? We provide, to our knowledge, the first systematic comparison of minimization proxies for LLM code refinement, empirically establishing that input token count is the most effective signal for guiding model reasoning.

## 3. Pilot Study: What Makes Feedback Effective?

Before presenting our full framework, we conduct a pilot study to empirically validate our central hypothesis that the quality of feedback, defined by its content and form, is more critical for successful code refinement than its quantity. This study is designed to isolate and examine these two fundamental dimensions by answering two questions: first, regarding **feedback content**, is property-oriented feedback, which conveys semantic rules, more effective than traditional I/O-based feedback? Second, concerning **feedback form**, is structurally minimal feedback, derived from the simplest possible counterexample, more effective at guiding an LLM's repair?

To ensure our findings are robust, our evaluation spans a diverse set of benchmarks, ranging from foundational function-level tasks (HumanEval (Chen et al., 2021) and MBPP (Austin et al., 2021), with their rigorous EvalPlus (Liu et al., 2024a) versions), to complex competition-level problems (LiveCodeBench (Jain et al., 2025)), and finally to real-world software issues (SWE-bench (Jimenez et al., 2024)). Our analysis is performed on three open-source LLMs representing varying capabilities: DeepSeek-Coder-V2 (Zhu et al., 2024), Qwen2.5-Coder (Hui et al., 2024), DeepSeek-R1 (Guo et al., 2025). Further details are provided in Appendix B.

*Table 1.* **Comparison of Different Feedback Minimization Strategies.** Results show the average pass@1 improvement over the baseline across three models for nine feedback selection strategies. The final column shows the average token cost per attempt on the DeepSeek-R1-Distilled-32B model. Token counts are calculated using each model's respective tokenizer.

| Strategy | | pass@1 (avg. of 3 models) | | | | | | Token Cost |
|---|---|---|---|---|---|---|---|---|
| Feature | Statistic | LCB-Easy | LCB-Mid | LCB-Hard | HumanEval | MBPP | SWE-bench | Avg. Tokens |
| Baseline | - | 76.0 | 36.4 | 10.5 | 85.8 | 65.3 | 16.6 | 4.73k |
| Line Coverage | Max | +8.9 | +5.2 | +2.1 | +6.1 | +7.3 | +3.3 | 5.53k |
| | Median | +9.5 | +5.3 | +3.0 | +6.3 | +7.2 | +3.2 | 5.12k |
| | Min | +12.4 | +5.8 | +3.7 | **+7.1** | +7.5 | **+3.7** | 4.87k |
| Runtime | Max | +8.6 | +5.3 | +2.1 | +6.1 | +7.2 | +3.2 | 5.76k |
| | Median | +11.2 | +5.3 | +3.5 | +6.6 | +7.4 | +3.4 | 5.10k |
| | Min | +11.4 | +5.6 | +3.8 | +7.0 | **+7.6** | +3.6 | 4.76k |
| Token Count | Max | +9.1 | +5.2 | +2.2 | +6.1 | +7.2 | +3.2 | 5.68k |
| | Median | +10.2 | +5.6 | +3.3 | +6.5 | +7.4 | +3.4 | 5.11k |
| | **Min** | **+13.2** | **+6.0** | **+4.0** | +7.0 | **+7.6** | **+3.7** | **4.72k** |

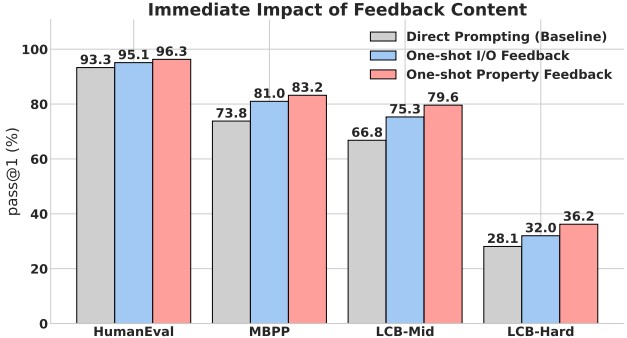

*Figure 2.* **The Impact of Feedback Content on One-shot Code Refinement.** We compare pass@1 rates of one-shot refinement using standard I/O feedback or property-oriented feedback.

### 3.1. Feedback Content

Our first investigation provides the foundational evidence for our property-oriented principle. We aim to answer a critical question: *Does framing a bug as a violation of a general semantic property, rather than a specific input-output (I/O) mismatch, improve an LLM's ability to correct it?* To isolate this variable, we design a one-shot code refinement experiment. For each program that initially fails a public test case, we create two distinct refinement scenarios:

- **Simple I/O Feedback.** The LLM is given the standard failing test case, including the input, the expected output, and the erroneous output it produced.
- **Property-Oriented Feedback.** The LLM is given feedback that reframes the exact same failure as a violation of a high-level program property.

We compare the pass@1 rates of these refinement attempts against the performance of the initial uncorrected baseline.

**Property-Oriented Feedback Drives More Effective Refinement.** As shown in Figure 2, the results reveal a clear and consistent performance hierarchy across all benchmarks.

*Table 2.* **Asymmetry of Verification.** Generation vs. verification accuracy on 100 LCB problems using DeepSeek-R1-32B. Detailed results and manual audit are provided in Appendix D.1.

| Difficulty | GenAcc. | VerAcc. | VerAcc. (w/ filtering) |
|---|---|---|---|
| Easy ($N = 32$) | 90.6% | 93.8% | 96.9% |
| Medium ($N = 34$) | 67.6% | 91.2% | 94.1% |
| Hard ($N = 34$) | 32.4% | 76.5% | 88.2% |
| **Overall** ($N = 100$) | **63.0%** | **87.0%** | **93.0%** |

Property-oriented feedback consistently outperforms simple I/O feedback, which in turn offers only a modest improvement over the unrefined baseline. This advantage is particularly pronounced on more challenging problems. For instance, on the LCB-Hard, standard I/O feedback provides a small boost, increasing pass@1 from a 28.1% baseline to 32.0%. However, by simply reframing the exact same error as a property violation, the pass@1 jumps to 36.2%, nearly doubling the performance gain. This finding suggests changing the feedback's content from a specific instance to a general rule allows the LLM to grasp the error's underlying semantic nature. This encourages the model to find a more generalizable solution that fixes the root cause, rather than merely patching the code to pass a single test case.

**Property-Oriented Feedback Mitigates Self-Deception.** Self-refinement often fails due to "cycle of self-deception," where LLMs repeat their own logical biases. Through validating high-level semantic rules rather than predicting specific test oracles, our property-oriented feedback mitigates this cycle via *asymmetry of verification* (Wei, 2025): verifying a solution's correctness is often a much easier task than generating that solution. Table 2 empirically confirm this principle. The accuracy of verification (87.0%) significantly outperforms code generation (63.0%), especially with a wider gap on hard problems (76.5% vs. 32.4%).

## 3.2. Feedback Form

Our second investigation studies the form of feedback once a property violation has been found. In software testing, failure-inducing inputs are often simplified to make faults easier to inspect (Misherghi & Su, 2006). For LLM-based refinement, however, the relevant notion of "simple" is not obvious: a counterexample may be short in tokens, execute few lines, or require little runtime, and these criteria need not rank examples in the same order. We therefore evaluate which simplicity proxy is most useful for selecting one counterexample as feedback. For each bug, we first construct a pool of property-violating counterexamples. We then choose one counterexample according to the minimum, median, or maximum value of each of the following metrics:

- **Line Coverage (Gopinath et al., 2014).** A classic software testing metric measuring the number of unique source code lines executed. Lower coverage corresponds to a shorter executed path through the candidate program.
- **Runtime.** A proxy for computational complexity, measured as execution time. Shorter runtime often corresponds to fewer loop iterations or simpler intermediate states.
- **Input Token Count.** A measure of the size of the counterexample as presented to the LLM. The input for each counterexample is tokenized using the tokenizer specific to the LLM being used for refinement.

This produces nine selection strategies. We apply each strategy in the same one-shot refinement setting and report the average pass@1 improvement across the tested models.

**Minimization Helps, and Token Count is the Strongest Proxy.** Table 1 shows two consistent trends in this setting. First, for each metric, selecting the minimum counterexample performs better than selecting the median or maximum counterexample. This supports the design choice of giving the model a small, focused failure rather than a larger one. Second, among the three minimum-based strategies, minimal **Token Count** gives the strongest overall pass@1 improvement and the lowest token cost. Line coverage and runtime capture properties of the program execution, while token count captures the amount of counterexample information included in the repair prompt. Based on this result, PGS selects feedback from the property-violating counterexample with the lowest input token count.

## 4. Property-Generated Solver

### 4.1. Problem Formulation

The primary objective in the code generation task is to employ an LLM to generate a program $C$ based on a given natural language specification $Q$ and a set of public (visible)

test cases $T_v$. Each test case $t_i = (I_i, O_i) \in T_v$ consists of an input $I_i$ and its corresponding expected output $O_i$. The quality of the generated program $C$ is ultimately evaluated against a set of private (hidden) test cases $T_h$. The program is considered correct if it passes all tests in $T_h$, *i.e.*, satisfying $\forall (I_j, O_j) \in T_h, C(I_j) = O_j$.

The core challenge is the generalization gap between public tests $T_v$ and private tests $T_h$. Since $T_v$ is often sparse, a generated program can easily overfit to the visible examples while still containing bugs that cause it to fail on $T_h$. While augmenting $T_v$ with auto-generated tests is a common remedy, it often compounds the issue by introducing significant feedback noise. Such auto-generated tests can inherit the model's own logical biases and struggle with the difficult test oracle problem, making them unreliable signals for correctness. Consequently, refinement relying on such limited, noisy feedback is inherently capped in effectiveness. The central task is thus to generate feedback that transcends these limitations, guiding LLMs toward generalizable and robust solutions.

### 4.2. Framework Workflow

As illustrated in Figure 3, we introduce the Property-Generated Solver (PGS), a framework that shifts the paradigm of TDD-based methods from relying on simple I/O feedback to utilizing property-oriented and structurally minimal feedback. PGS establishes a collaborative workflow between two specialized LLM-based agents, *i.e.*, a **Generator** and a **Tester**. Both agents can be implemented using general-purpose LLMs such as ChatGPT (OpenAI, 2023) or DeepSeek-R1 (Guo et al., 2025). The Generator performs code generation and refinement, while the Tester challenges the generated code by validating it against high-level properties and formulates minimal and highly actionable feedback based on property violation. The specific prompt templates are detailed in Appendix A.

**Initial Code Generation.** The process begins with the Generator producing an initial version of the program $C$ based on the problem specification $Q$. This serves as the starting point for the iterative refinement cycle.

**Property Definition.** Following initial code generation, the Tester agent defines a set of candidate properties, $\mathcal{P}$, based on the natural language description $Q$. These properties are high-level semantic rules, ranging from global invariants (*e.g.*, "the output list must be sorted") to partial specifications (*e.g.*, "all elements in the output must be prime"). The Tester then translates these abstract properties into executable checking code, $C^{\mathcal{P}}$, typically structured as assertion statements or boolean-valued verification functions. To ensure the soundness of LLM-generated properties, each property check in $C^{\mathcal{P}}$ is validated against the public test cases $T_v$. This critical filtering process discards any property that contradicts the known ground truth, ensuring that the feedback provided to the Generator is based on reliable

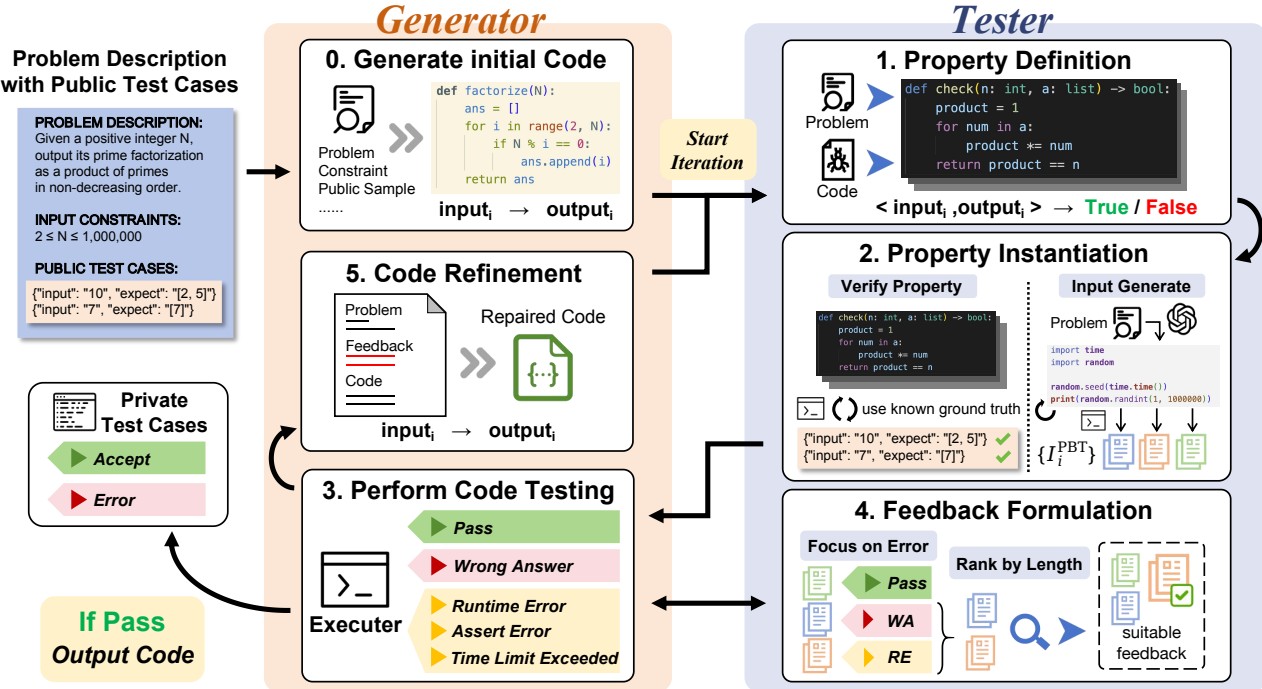

*Figure 3.* **Overview of the Property-Generated Solver (PGS) Framework.** PGS establishes a collaborative workflow between the Generator and the Tester. **(0)** The Generator creates an initial program from the problem. **(1)** The Tester defines high-level properties and translates them into executable checks. **(2)** The Tester validates these properties and synthesizes a diverse pool of inputs for probing. **(3)** The program is executed, with property violations being identified. **(4)** The Tester analyzes property violations and formulates feedback from the minimal counterexample. **(5)** The Generator uses this targeted feedback to refine the code. The cycle (steps 1-5) then repeats.

and correct semantic rules.

**Input Synthesis for Property Probing.** Once sound property checks $C^{\mathcal{P}}$ is established, the Tester generates a diverse pool of inputs, $\{I^{\mathcal{P}}\}$, to probe for violations. A key challenge is creating inputs semantically rich enough to stress-test the program logic, where simple random or fuzzing techniques often prove insufficient as they tend to explore shallow logic flow. To overcome this, we adopt an LLM-driven synthesis technique (El-Kishky et al., 2025). We prompt the Generator model, using only the problem description and constraints, to act as a dedicated "test input generator" that creates a varied set of potential counterexamples. These synthesized inputs $\{I^{\mathcal{P}}\}$ are then used along with $C^{\mathcal{P}}$ in the subsequent validation step, enabling PGS to uncover bugs that simple I/O checks would miss.

**Property-Driven Validation.** For validation, the Tester instruments the candidate program $C$ by injecting the property checks $C^{\mathcal{P}}$ directly into its source code, creating a unified executable $C'$. By treating properties as an intrinsic part of the program's logic, PGS transforms latent semantic bugs, which might otherwise result in a silent "Wrong Answer", into explicit, machine-checkable runtime errors like an *AssertionError*. This instrumentation is crucial, since it makes previously hidden bugs tractable, and further encourages the LLM to reason about the code and the problem specifications holistically. The instrumented program $C'$ is then

executed against the full suite of inputs, including public test cases $\boldsymbol{T}_v$ and the synthesized inputs $\{I^{\mathcal{P}}\}$, to collect all outcomes for the next stage.

**Feedback Formulation.** A buggy program can fail on numerous inputs, but providing all raw outcomes would create a noisy and overwhelming signal. Inspired by delta-debugging principles (Misherghi & Su, 2006) and our pilot study, the Tester's role is to distill this information into a single, potent piece of feedback. From the set of all property-violating inputs, the Tester strategically selects the counterexample with the minimal input token count (a strategy empirically validated in Section 3.2). This structurally minimal feedback isolates the error's root cause, removes distracting noise, and provides the clear, actionable signal necessary for effective LLM-driven refinement.

**Code Refinement.** The Generator receives the property-oriented and structurally minimal feedback, which comprises the failing input, the program's erroneous output, and the specific violated property. This structured information is used to construct a new prompt that instructs the Generator to analyze the failure and produce a corrected version of the code. The iterative refinement cycle continues until the program passes all property checks and public test cases, or until a predefined budget is exhausted (*e.g.*, a maximum of five refinement attempts), steering the LLM toward a more robust and functionally correct solution.

*Table 3.* Comparison on Code Generation across Multiple Benchmarks. We report pass@1 scores with standard deviations. "DS" denotes DeepSeek, and "Claude-4" denotes Claude-Sonnet-4. The best result in each row is highlighted in **bold**. Additional baseline results and Fix Rate are provided in Appendix B (Table 6).

| Dataset | Method | DS-V2 | Qwen2.5 | DS-R1-32B | Qwen3-30B | DS-V3.1 | Claude-4 |
|---|---|---|---|---|---|---|---|
| **HumanEval (HE)** | Baseline | $76.2 \pm 0.7$ | $87.8 \pm 0.6$ | $93.3 \pm 0.5$ | $91.5 \pm 0.8$ | $95.5 \pm 0.7$ | $97.2 \pm 0.6$ |
| | Code-T | $81.1 \pm 1.7$ | $88.4 \pm 1.5$ | $94.5 \pm 1.4$ | $92.2 \pm 0.6$ | $96.2 \pm 0.5$ | $97.8 \pm 0.4$ |
| | Self-Edit | $81.7 \pm 1.6$ | $90.2 \pm 1.7$ | $95.1 \pm 1.6$ | $92.8 \pm 0.7$ | $96.5 \pm 0.6$ | $97.9 \pm 0.5$ |
| | MGDebugger | $83.5 \pm 1.7$ | $92.1 \pm 1.6$ | $95.7 \pm 1.5$ | $93.2 \pm 0.8$ | $96.8 \pm 0.7$ | $98.2 \pm 0.6$ |
| | Self-Debugging | $84.1 \pm 1.9$ | $92.7 \pm 1.8$ | $96.3 \pm 1.7$ | $93.5 \pm 0.9$ | $97.1 \pm 0.8$ | $98.5 \pm 0.7$ |
| | Reflexion | $86.6 \pm 1.2$ | $91.5 \pm 1.4$ | $95.1 \pm 1.3$ | $92.9 \pm 0.6$ | $96.9 \pm 0.8$ | $98.5 \pm 0.6$ |
| | **PGS (Ours)** | $\mathbf{89.0 \pm 1.5}$ | $\mathbf{94.5 \pm 1.1}$ | $\mathbf{97.6 \pm 1.0}$ | $\mathbf{95.2 \pm 0.8}$ | $\mathbf{98.2 \pm 0.6}$ | $\mathbf{99.1 \pm 0.3}$ |
| **MBPP** | Baseline | $56.8 \pm 0.6$ | $65.4 \pm 0.7$ | $73.8 \pm 0.6$ | $73.1 \pm 0.9$ | $89.5 \pm 0.8$ | $93.5 \pm 0.3$ |
| | Code-T | $60.4 \pm 1.5$ | $69.4 \pm 1.6$ | $82.4 \pm 1.0$ | $76.5 \pm 1.2$ | $90.8 \pm 1.1$ | $94.5 \pm 0.5$ |
| | Self-Edit | $62.4 \pm 1.5$ | $70.2 \pm 1.8$ | $83.0 \pm 1.2$ | $78.2 \pm 1.3$ | $91.2 \pm 1.2$ | $94.8 \pm 0.6$ |
| | MGDebugger | $63.8 \pm 2.0$ | $71.2 \pm 1.9$ | $83.8 \pm 1.3$ | $79.6 \pm 1.4$ | $91.5 \pm 1.3$ | $95.2 \pm 0.7$ |
| | Self-Debugging | $63.8 \pm 1.9$ | $72.4 \pm 2.0$ | $84.4 \pm 1.4$ | $80.0 \pm 1.5$ | $91.8 \pm 1.4$ | $95.5 \pm 0.8$ |
| | **PGS (Ours)** | $\mathbf{67.6 \pm 1.8}$ | $\mathbf{76.6 \pm 1.9}$ | $\mathbf{87.2 \pm 1.3}$ | $\mathbf{82.5 \pm 1.5}$ | $\mathbf{94.1 \pm 1.4}$ | $\mathbf{96.5 \pm 0.8}$ |
| **LiveCodeBench (LCB)** | Baseline | $26.7 \pm 0.8$ | $31.8 \pm 0.9$ | $64.4 \pm 0.9$ | $52.2 \pm 1.0$ | $72.5 \pm 2.6$ | $63.1 \pm 1.7$ |
| | Code-T | $29.2 \pm 1.3$ | $34.6 \pm 1.4$ | $70.8 \pm 1.6$ | $54.5 \pm 1.8$ | $75.5 \pm 2.7$ | $68.2 \pm 1.4$ |
| | Self-Edit | $30.2 \pm 1.9$ | $35.2 \pm 1.8$ | $73.6 \pm 1.8$ | $60.2 \pm 2.0$ | $79.8 \pm 2.8$ | $70.5 \pm 1.9$ |
| | Self-Debugging | $31.3 \pm 2.2$ | $38.5 \pm 2.0$ | $72.5 \pm 2.0$ | $61.5 \pm 2.1$ | $80.5 \pm 2.0$ | $72.8 \pm 2.0$ |
| | **PGS (Ours)** | $\mathbf{34.1 \pm 1.7}$ | $\mathbf{40.0 \pm 1.9}$ | $\mathbf{76.5 \pm 1.8}$ | $\mathbf{65.1 \pm 1.4}$ | $\mathbf{83.2 \pm 2.0}$ | $\mathbf{75.5 \pm 1.8}$ |
| **CodeContests (CC)** | Baseline | $12.5 \pm 0.6$ | $14.4 \pm 0.7$ | $38.1 \pm 0.8$ | $30.8 \pm 0.9$ | $46.8 \pm 1.1$ | $42.1 \pm 1.0$ |
| | Code-T | $14.2 \pm 0.9$ | $15.9 \pm 1.0$ | $42.9 \pm 1.4$ | $33.2 \pm 1.6$ | $49.6 \pm 1.4$ | $44.3 \pm 1.5$ |
| | Self-Edit | $15.6 \pm 1.7$ | $16.4 \pm 1.5$ | $44.8 \pm 1.6$ | $34.4 \pm 1.7$ | $53.2 \pm 1.6$ | $48.5 \pm 1.6$ |
| | Self-Debugging | $16.1 \pm 2.1$ | $17.3 \pm 1.9$ | $45.8 \pm 1.9$ | $36.1 \pm 2.0$ | $54.5 \pm 1.8$ | $49.9 \pm 1.9$ |
| | **PGS (Ours)** | $\mathbf{20.2 \pm 1.8}$ | $\mathbf{22.4 \pm 2.0}$ | $\mathbf{49.7 \pm 2.0}$ | $\mathbf{41.7 \pm 2.2}$ | $\mathbf{60.2 \pm 2.1}$ | $\mathbf{55.9 \pm 2.1}$ |
| **SWE-bench** | SWE-agent | $9.8 \pm 1.4$ | $10.3 \pm 0.8$ | $34.4 \pm 2.0$ | $46.5 \pm 1.7$ | $54.2 \pm 1.8$ | $65.5 \pm 1.6$ |
| | **PGS (Ours)** | $\mathbf{11.9 \pm 1.0}$ | $\mathbf{12.8 \pm 0.5}$ | $\mathbf{37.3 \pm 2.3}$ | $\mathbf{50.7 \pm 1.9}$ | $\mathbf{58.4 \pm 2.5}$ | $\mathbf{70.2 \pm 1.5}$ |

## 5. Experiment

### 5.1. Experimental Setup

**Comparison Counterparts.** We evaluate PGS against multiple state-of-the-art counterparts in code refinement, including: (1) non-iterative methods (Direct or CoT prompting (Wei et al., 2022)); (2) TDD frameworks that explicitly use test cases, such as Code-T (Chen et al., 2023a) and Reflexion (Shinn et al., 2024); (3) sophisticated self-correction frameworks that emulate debugging, including Self-Edit (Zhang et al., 2023), Self-Debugging (Chen et al., 2023b), MGDebugger (Shi et al., 2024), and LDB (Zhong et al., 2024). To ensure a fair comparison, all iterative methods were executed under a matched computational budget, capped at 5 refinement attempts, identical to PGS.

**Benchmarks and Foundation Models.** To assess the generalizability of our approach, we evaluate on five widely-recognized benchmarks that span a spectrum of tasks from function-level synthesis to real-world software engineering: HumanEval (Chen et al., 2021), MBPP (Austin et al., 2021), LiveCodeBench (Jain et al., 2025), CodeContests (Li et al., 2022), and SWE-bench (Jimenez et al., 2024). Our evaluation employs six LLMs across a broad spectrum of capabilities: from open-source task-specific models like DeepSeek-

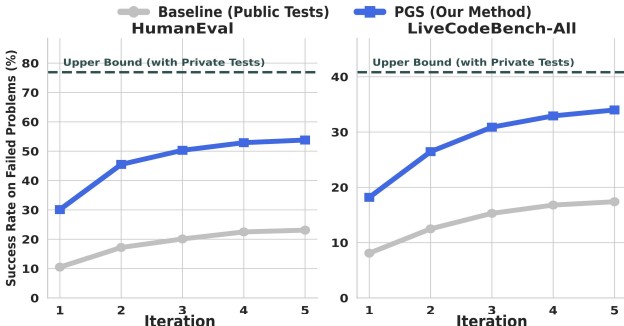

*Figure 4.* Iterative fix rate of PGS on initially-failed problems from **HumanEval** (with DS-V2) and **LiveCodeBench** (with DS-R1-32B). The dashed line indicates the performance upper bound.

Coder-V2 (Zhu et al., 2024), Qwen2.5-Coder (Hui et al., 2024), to advanced reasoning and proprietary models including DeepSeek-R1, Qwen3 (Yang et al., 2025a), DeepSeek-V3.1 and Claude-Sonnet-4. To ensure robustness, we report average pass@1 scores across 5 independent runs.

**Implementation Details.** PGS runs for up to 5 iterations, synthesizing 5 candidate properties and 64 test inputs per round. Based on our pilot study (Section 3), we use property violations from counterexamples with the minimum input token count. More details can be found in Appendix B.

## 5.2. Main Result

As shown in Table 3, the PGS consistently establishes a new state-of-the-art for code refinement, substantially outperforming other methods across every benchmark and LLMs. This demonstrates the model-agnostic benefits of our high-quality feedback approach. This advantage is particularly pronounced on the challenging LiveCodeBench benchmark, where PGS with DeepSeek-R1-32B achieves a 12.1% improvement over the baseline, far surpassing other iterative methods. Besides, PGS outperforms sophisticated debugging frameworks. On HumanEval, for instance, PGS with DeepSeek-Coder-V2 reaches 89.0% pass@1, a significant lead over methods like MGDebugger (83.5%). While these methods excel at tracing execution for given tests, their scope is fundamentally limited by those initial tests. PGS, by contrast, actively probes the code's semantic boundaries, allowing it to uncover and repair a wider class of bugs.

## 5.3. Ablation Study

We evaluate the effectiveness of PGS's core design through a series of analyses, including input synthesis impact, cost-performance trade-offs, and cross-model robustness. These experiments validate our strategy for mitigating the self-deception cycle and transforming latent bugs into actionable signals. More ablations are provided in Appendix D.

### 5.3.1. CONTRIBUTION OF EACH COMPONENT

To isolate the contribution of each component in PGS, we conduct an ablation study on LiveCodeBench using DeepSeek-R1-Distilled-32B. As shown in Table 4, structural minimization improves the raw I/O feedback baseline from 64.4% to 67.2%, confirming that concise failure signals help refinement even without property feedback. Property generation further improves pass@1 to 68.5%, and adding synthesized inputs increases it to 69.4%, showing that additional counterexamples expose errors beyond public tests. Property filtering provides a larger gain, reaching 71.6%, which indicates the importance of removing invalid or contradictory properties before using them as feedback. Finally, iterative refinement yields the full PGS result of 76.5%, showing that multi-round correction is important for complex failures.

### 5.3.2. EFFECTIVENESS OF INPUT SYNTHESIS

PGS relies on the Tester agent's ability to synthesize inputs that probe property violations. To quantify this, we analyze the most challenging cases: where the initial code fails in hidden test cases. We compare PGS against a baseline TDD approach and an oracle ceiling representing feedback from all hidden private tests.

Figure 4 demonstrate the advantage of synthesis-driven refinement. On HumanEval and LiveCodeBench, PGS

*Table 4.* **Contribution of Each Component in PGS.**

| Method / Component | pass@1 (%) | Δ |
|---|---|---|
| **Baseline** (Raw I/O Feedback) | 64.4 | - |
| + Structural Minimization | 67.2 | +2.8 |
| + Property Generation | 68.5 | +1.3 |
| + Input Synthesis | 69.4 | +0.9 |
| + Property Check (Filtering) | 71.6 | +2.2 |
| + Iterative Refinement (**Full PGS**) | **76.5** | +4.9 |

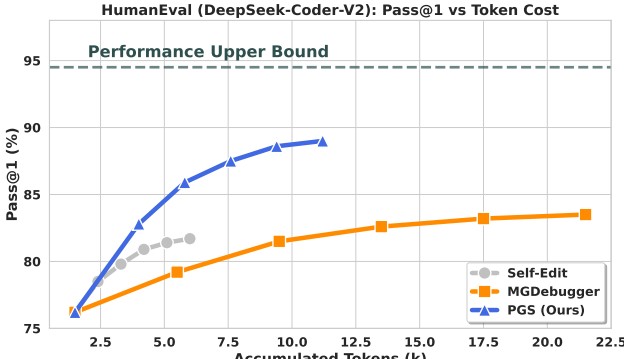

*Figure 5.* **pass@1 performance *vs.* Token Consumption.** Token counts are calculated using each model's respective tokenizer.

achieves higher fix rates faster than the baseline, with the gap widening per iteration. On HumanEval, the baseline plateaus at 23.1%, while PGS reaches 53.8%. Given the 76.9% oracle ceiling, PGS recovers 57% of the gap between baseline and oracle. This trend holds on LiveCodeBench, where PGS closes 71% of the gap between the baseline (17.4%) and oracle (40.8%). These results prove that synthesized inputs are not redundant; they effectively uncover deep logical flaws typically caught only by private tests, validating our overall strategy.

Appendix D further analyzes input-source variants, including same-model synthesis, cross-model synthesis, and hidden inputs without labels (Table 8). This ablation checks whether the benefit of input synthesis depends on using the same model that performs code generation. The near-identical performance of same-model and cross-model synthesis suggests that the gains mainly come from executable property checks over additional inputs, rather than from model-specific self-bias in the synthesis step.

### 5.3.3. COST-EFFECTIVENESS ANALYSIS

We analyze the cost-effectiveness trade-off between token consumption and pass@1 over 5 iterations. We compare against Self-Edit and MGDebugger because they represent two representative refinement regimes: lightweight self-editing with raw execution feedback and more expensive multi-level debugging with richer diagnostic traces. When compared with simple baselines like Self-Edit, although

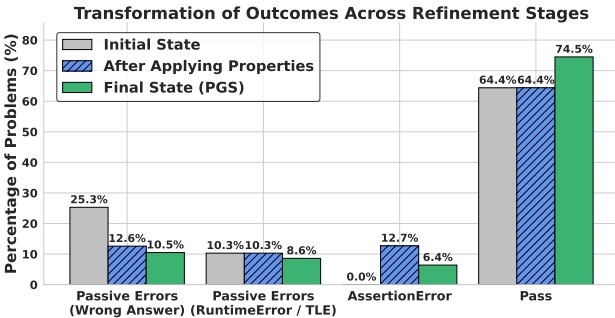

*Figure 6.* Ambiguous "Wrong Answer" failures are converted into explicit "AssertionError"s, which are then resolved, increasing the final "Pass" rate.

*Table 5.* PGS Performance across Heterogeneous Generator-Tester Configurations. More results are provided in Appendix D.3

| Generator | Tester | HE | MBPP | LCB | CC |
|-----------|--------|------|------|------|------|
| DS-R1-32B | DS-R1-32B | 97.6 | 87.2 | 76.5 | 49.7 |
| DS-R1-32B | Qwen2.5 | 96.8 | 86.6 | 74.9 | 48.4 |
| DS-R1-32B | Qwen3-30B | 97.2 | 87.2 | 76.1 | 49.6 |

PGS consumes more tokens ($1.6\times$) per iteration due to the property generation step, this cost is quickly offset by its superior performance gains. As illustrated in Fig 5, PGS after only 2 iterations already surpasses the final performance of Self-Edit after 5 full iterations with lower token costs. When compared with complex multi-level debugging baselines like MGDebugger, PGS is significantly more efficient in both pass@1 and token costs. In summary, PGS offers the optimal trade-off, delivering high accuracy with lower computational overhead than its counterparts. Additional multi-dataset results with average token cost per problem are provided in Appendix D.4.

### 5.3.4. MITIGATING THE CYCLE OF SELF-DECEPTION

To examine whether PGS mitigates the cycle of self-deception, where a model fails to recognize its own logical flaws, we evaluate the framework across diverse Generator-Tester pairs. This setup separates code generation from feedback construction, reducing the chance that the same model simply confirms its own erroneous reasoning. As shown in Table 5 and Appendix D.3, PGS maintains consistent gains across different model families. Notably, PGS remains robust in asymmetric configurations: for instance, using the smaller Qwen2.5 as a Tester still significantly boosts the more powerful DeepSeek-R1-32B. This confirms that property validation is simpler than code synthesis, allowing a smaller agent to provide reliable guidance. Since the resulting feedback is grounded in executable property checks rather than LLM self-evaluation, the gains are less likely to come from model-specific preference biases. These results, supported by Appendix D.1, suggest that PGS mitigates the self-deception trap through independent semantic validation.

### 5.3.5. LATENT BUGS AS ACTIONABLE SIGNALS

We now decompose the underlying mechanism of PGS into three distinct stages to analyze its potency: **(1) Initial** baseline state; **(2) Property Injection**, where checking code is instrumented; and **(3) Final** refined state.

As shown in Figure 6, the transition from Stage 1 to 2 reveals the core decomposition. While the **Pass** rate remains 64.4%, **Wrong Answer** outcomes are nearly halved from **25.3%** to **12.6%**. This reduction is almost perfectly mirrored by the emergence of **AssertionError**, indicating a shift in observability rather than logic. Injected properties act as a diagnostic lens, surfacing latent bugs by converting vague semantic errors into explicit violations.

This transformation is critical for subsequent refinement. In Stage 3, the Generator leverages these **AssertionError** signals to debug the code, reducing the category to just **6.4%**. This effective resolution drives the final **Pass** rate to **74.5%**. This analysis confirms that PGS operates via a two-step principle: first, making hidden bugs visible through injection, and second, leveraging these signals for precise refinement. Detailed case studies are available in Appendix C.

## 6. Limitations

PGS relies on LLM-generated properties to construct semantic feedback. Although public-test filtering removes properties that contradict visible examples, it does not provide a formal guarantee of correctness or coverage; a property may still miss edge cases or impose constraints that are too strong. This risk is more pronounced for repository-level bugs or tasks whose intended behavior depends on implicit API contracts. PGS also introduces additional token, runtime, and execution overhead because it generates properties, synthesizes probing inputs, and performs iterative checks. Finally, passing PGS-generated checks should not be treated as full verification, since a repair may overfit to the current properties or fail when the decisive fault is not covered. These limitations motivate future work on stronger property validation, broader property coverage, and independent final validation for repaired code.

## 7. Conclusion

We improve LLM code correctness by establishing that effective refinement requires property-oriented, structurally minimal feedback. This principle informs the design of our PBT-based framework, PGS, which achieves SOTA performance by converting latent bugs into explicit repair signals. By converting latent bugs into explicit, minimal refinement signals, PGS prioritizes feedback quality over test quantity. Our work offers a robust methodology for reliable code generation, with future efforts focused on enhancing autonomous property discovery.

## Acknowledgements

This work was supported by National Natural Science Foundation of China (62132001), Beijing Natural Science Foundation (L252218), and the Fundamental Research Funds for the Central Universities. This work is partially supported by Shanghai Innovation Institute.

## Impact Statement

This paper presents work whose goal is to advance the field of Machine Learning. There are many potential societal consequences of our work, none which we feel must be specifically highlighted here.

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

# A. Property-Generated Solver Settings

## A.1. Category of Code Execution Outcomes

In the main body of the paper, we discussed that our framework's effectiveness stems from generating high-quality, targeted feedback. A cornerstone of this strategy is the ability to differentiate between various outcomes of code execution. Each outcome category provides a distinct type of signal about the nature of the flaw in the generated code, allowing our Tester agent to formulate the most effective and actionable feedback for the Generator agent. Below, we detail these categories and the corresponding feedback strategy employed by PGS.

**Pass.** This is the successful outcome, where the candidate code correctly passes a given test case (either a public test or a property-driven check). No corrective feedback is needed for this instance. If the code passes all available public tests and all synthesized property checks, the refinement process concludes successfully.

**Property Violation (`AssertionError`).** This is the primary signal leveraged by the PGS framework and represents the most informative type of failure. A property violation occurs when the code executes without crashing but fails a custom-injected assertion that encodes a high-level program invariant (e.g., a sorted list must be non-decreasing). This mechanism is crucial for converting *latent logical flaws*—bugs that might not be caught by simple input-output checks—into explicit, machine-checkable failures. **Feedback Strategy:** This outcome receives the highest priority. The Tester provides feedback centered on the minimal counterexample that triggered the `AssertionError`, along with a description of the violated property. This gives the Generator a clear, semantic-level clue about what is wrong with its logic.

**Wrong Answer (WA).** This outcome occurs when the program's output does not match the expected output for a known, ground-truth test case (typically the public examples provided with the problem). This indicates a fundamental logical error. **Feedback Strategy:** The feedback consists of the input, the model's incorrect output, and the correct, expected output. This is the classic feedback loop in Test-Driven Development (TDD), and PGS uses it to correct basic functional errors.

**Runtime Error (RE).** This category includes errors that cause the program to crash during execution, such as `IndexError`, `TypeError`, or `ZeroDivisionError`. These errors often point to faulty assumptions about the input data or incorrect state management. **Feedback Strategy:** The feedback includes the full error message and stack trace, along with the specific input that caused the crash. This provides the Generator with a precise pointer to the location and type of the bug in the code.

**Time Limit Exceeded (TLE).** This outcome signifies that the generated code is correct in its logic but is too slow to pass within the allocated time constraints. This points to a flaw in algorithmic efficiency rather than logical correctness. **Feedback Strategy:** The feedback highlights the inefficiency. It provides the input that caused the TLE and explicitly prompts the Generator to rethink its algorithmic approach to find a more time-efficient solution. This guides the model towards high-level algorithmic optimization.

## A.2. Prompt Templates

This section provides the detailed prompt templates used throughout our Property-Generated Solver (PGS) framework. These prompts are designed to guide the Generator and Tester agents through the key stages of the code refinement process. The prompts are designed to be clear and role-specific, ensuring that each agent performs its designated task effectively.

Each prompt is tailored for a specific stage of the refinement process, guiding our agents to perform their tasks. The process begins with the **Initial Code Generation** prompt, which instructs the Generator to produce a baseline solution (Table 12). Concurrently, the Tester agent uses the **Property Generation** prompt to distill high-level program properties (Table 13). Subsequently, the Tester employs the **Input Synthesis** prompt to generate diverse inputs for probing these properties (Table 14). Finally, if a violation is detected, the cycle culminates in the **Feedback-Driven Repair** prompt, which provides the Generator with a minimal counterexample to fix the bug (Table 15).

# B. More Experimental Setup

This section details the experimental setup, including the hardware and software environment, as well as the specific parameters used for model inference during our experiments.

**Benchmarks.** Following prior works (Zhang et al., 2024a; Shi et al., 2024; Chen et al., 2023b), our evaluation utilizes three prominent code generation benchmarks:

- **HumanEval** (Chen et al., 2021). A standard benchmark comprising 164 handwritten Python programming problems designed to evaluate the function-level code synthesis capabilities of LLMs. During the generation and refinement process, models are provided with the problem description and any canonical tests accompanying the original HumanEval problem statements. Final validation is performed using the benchmark's standard hidden test cases.

- **MBPP** (Austin et al., 2021). This benchmark consists of approximately 500 crowd-sourced entry-level Python programming problems. The models receive the problem description and the first hidden test case during the generation phase (Ni et al., 2023). Final validation is performed using the benchmark's standard hidden test cases.

- **LiveCodeBench** (Jain et al., 2025). A challenging benchmark featuring problems sourced from live programming contests, often requiring more complex algorithmic reasoning, intricate I/O handling, and adherence to stricter execution constraints. To ensure a comprehensive and up-to-date evaluation, we utilize the latest "v5" version, comprising 880 problems. For all problems from this benchmark, the public test cases provided with each problem description are made available to all Test-Driven Development methods, including PGS and relevant baselines.

- **CodeContests** (Li et al., 2022). A highly competitive benchmark that assesses advanced problem-solving and algorithmic implementation skills. While the full dataset features thousands of problems from various competitions, our evaluation utilizes the official `valid` split, which consists of 117 problems. During the generation and refinement process, models are given access to the problem description along with the public (sample) test cases. Final validation is performed against a comprehensive set of hidden test cases, demanding solutions that are not only correct but also efficient enough to pass strict time and memory limits.

- **SWE-bench** (Jimenez et al., 2024). A rigorous and realistic benchmark designed to evaluate LLMs on genuine software engineering tasks. Our evaluation focuses on the challenging "Verified" split, which comprises 500 real-world bug-fixing scenarios extracted from popular GitHub repositories. For each task, models are provided with the full code repository, the problematic GitHub issue description, and any associated pull request information. Final validation is performed by executing the project's original test suite against the generated patch, requiring the fix to resolve the issue without introducing any regressions.

**Comparison Baselines.** We compare PGS against a carefully selected suite of baselines that represent the key paradigms in state-of-the-art code refinement. We categorize them as follows:

- **Foundational Baselines.** These methods measure the raw capabilities of the LLMs.
  - *Direct & CoT Prompting.* We evaluate both zero-shot baseline and Chain-of-Thought (CoT) (Wei et al., 2022) prompting to establish the fundamental performance ceiling without iterative refinement.

- **Test-Driven Refinement Frameworks.** These methods explicitly use test outcomes to guide the refinement process, representing the direct alternative to our approach.
  - *Code-T* (Chen et al., 2023a). Leverages auto-generated tests to rank code candidates.
  - *Reflexion* (Shinn et al., 2024). A method that uses self-reflection to improve code.

- **Self-Correction and Debugging Frameworks.** These are more sophisticated methods that emulate a debugging process to identify and fix errors.
  - *Self-Edit* (Zhang et al., 2023). A technique where the LLM attempts to refine its own code based on execution feedback.
  - *Self-Debugging* (Chen et al., 2023b). An iterative method where the LLM explains its code and simulates a debugging process to fix bugs.
  - *MGDebugger* (Shi et al., 2024). A multi-level framework that identifies and fixes errors at different levels of code abstraction.
  - *LDB* (Zhong et al., 2024). A refinement technique that tracks intermediate variable values during runtime to locate and repair errors.

- *SWE-agent* (Yang et al., 2024): An open source framework for repository-level code refinement, which uses a ReAct loop with specialized tools for navigating and editing codebases. We include it as a primary baseline for SWE-bench experiments.

All methods are provided with identical problem descriptions and public test cases to ensure a fair comparison. Private test cases are strictly withheld during the iterative loops and are used solely for final evaluation.

**Foundation Models.** We select three LLMs with different capabilities to implement proposed PGS. Based on their general coding proficiency, they are listed from weak to strong as follows:

- **DeepSeek-Coder-V2 (Zhu et al., 2024).** A powerful open-source model specifically optimized for code generation tasks.

- **Qwen2.5-Coder (Hui et al., 2024).** A strong open-source model from the Qwen series, known for its advanced coding abilities.

- **DeepSeek-R1-Distilled-32B (Guo et al., 2025).** A highly capable LLM featured with long CoT reasoning. We utilize a variant 32B distilled model, which aims to offer a strong balance of performance and efficiency.

For all models, we follow official configurations (*e.g.*, maximum context window of tokens, temperature, specific version identifiers) to guarantee a consistent setup.

**Metrics.** We adopt two metrics to evaluate the effectiveness of PGS:

- **pass@1 (Yu et al., 2024)** measures the overall proportion of problems for which the generated final code successfully passes all hidden (private) test cases.

- **Fix Rate (Repair Success Rate) (Yasunaga & Liang, 2021)** quantifies the proportion of initially incorrect code samples that are successfully corrected by the iterative refinement process to pass all hidden test cases.

**Tool Usage and Grounding.** PGS employs different tooling strategies based on task complexity.

- **Function-level tasks (e.g., HumanEval, MBPP)**: We use lightweight execution environments for Python code validation.

- **Competition-level tasks (e.g., LiveCodeBench, CodeContests)**: We incorporate problem specific I/O handlers and time constraints.

- **Repository-level tasks (e.g., SWE-bench)**: To ensure fair comparison, we adopt the same toolset and grounding as SWE-agent, including bash terminal, file navigation, edit commands, and test execution.

**Hardware and Software Environment.** All experiments were conducted on a server equipped with **4x NVIDIA H100 GPUs**. The operating system environment was configured with **Python 3.11** and **CUDA 12.2**. Our software stack was built upon **PyTorch 2.6**, and we utilized the **vLLM library (version 0.8.5.post1)** for efficient and high-throughput inference of all large language models. To maximize performance, we configured vLLM to use a **4-way tensor parallelism** strategy across the available GPUs.

**Inference Parameters.** For all tasks, we use a temperature of 0.7 and nucleus sampling with `top_p=0.95` to balance creativity and coherence, following DeepSeek-R1 and Qwen-2.5-Coder technical reports. The `max_tokens` is set to 32,768 to prevent premature truncation during complex reasoning. All token counts for structural minimization are calculated using each model's respective tokenizer to ensure precision across heterogeneous architectures.

**Baseline Consistency.** To ensure a fair and rigorous comparison, all baseline methods evaluated in our study were executed using the identical hardware setup and inference configuration described above. Specifically, they shared the same sampling strategy (`top_p=0.95`) and `max_tokens` setting. This consistency eliminates variability due to different generation

parameters and ensures that observed performance differences can be attributed to the core methodologies of the frameworks themselves.

**Comprehensive Experimental Results**

While the main text focuses on the most representative baselines for conciseness, we conducted a broader set of evaluations against additional competitive methods. As shown in Table 6, these extended results offer a more comprehensive view of PGS's performance relative to the current state-of-the-art in automated code refinement.

## C. Case Studies

To provide a concrete, step-by-step illustration of our Property-Generated Solver (PGS) framework in action, we present two detailed case studies. These cases are chosen to demonstrate how the core principles discussed in the main body—property-oriented feedback and structural minimization—are operationalized to resolve different types of challenging bugs. Each case follows the full refinement workflow, from a flawed initial code to a correct final version, highlighting how targeted feedback drives effective debugging.

The first case study, presented in Figure 8, showcases our framework's primary mechanism for **transforming latent logical flaws into explicit, actionable signals**. It begins with an initial solution that is overly complex and contains a subtle bug related to unhandled states (resulting in an infinite cost). We show how injecting simple, property-based assertions (e.g., 'cost must be finite') allows a minimal counterexample ("0011") to trigger a clear `AssertionError`. This precise, low-noise signal pinpoints the exact logical failure, guiding the LLM to produce a final solution that is not only correct but also significantly more concise and elegant.

The second case study, detailed in Figure 9, demonstrates PGS's ability to **invalidate an entire algorithmic approach by using a simpler, trusted implementation as a property**. Here, the initial code attempts a sophisticated optimization using cycle detection, which contains a flaw. The key property injected is that for a subset of simple inputs (small 'k'), the output of the complex algorithm must match that of a direct, brute-force simulation. The simulation acts as a temporary "oracle." The resulting `AssertionError` explicitly shows a discrepancy between the optimization and the trusted simulation. This potent feedback prompts the model to abandon the flawed approach entirely and converge on a robust, correct solution using a different paradigm (dynamic programming with binary lifting), a feat difficult to achieve with traditional feedback.

## D. More Ablation Study

This section dissects the core mechanisms of PGS through comprehensive ablation and empirical analysis. We extend the primary evaluation by investigating: (1) the **asymmetry of verification** as a foundation for the Tester-Generator roles; (2) the **input source** used for property probing; (3) the **generalizability** across C++ and **cross-model configurations** to address model-specific biases; and (4) an extended **cost-effectiveness analysis** Collectively, these analyses substantiate that PGS is a robust, model-agnostic, and resource-efficient strategy for automated code refinement.

### D.1. Asymmetry of Verification

"Cycle of self-deception" is a fundamental challenge in LLM-based self-correction, where the LLM shares the same logical biases when it generates the code or predicts an oracle. Our PGS framework is designed to mitigate this cycle by avoiding oracle prediction and instead leveraging a core principle: the **Asymmetry of Verification** (Wei, 2025), *i.e.*, verifying a solution's correctness is often a significantly easier task than generating that solution from scratch.

**Roles of Generator and Tester in PGS.** The core design of PGS adheres to this principle. The Generator in PGS is responsible for producing complex algorithmic logic (*e.g.*, a complete prime factorization algorithm), while the Tester only needs to generate several simple, independent property checks (*e.g.*, `assert math.prod(factors) == N`). The risk of an LLM failing to generate a simple assertion (verification) is significantly lower than its risk of failing to generate a complex algorithm (generation), thus ensuring that the Tester can provide reliable and valid guidance even when the Generator fails.

**Empirical Evidence.** To provide rigorous empirical evidence for this asymmetry, we conduct an additional experiment on a 100-problem subset of LiveCodeBench, which consists of 32 Easy, 34 Medium, and 34 Hard problems, using DeepSeek-R1-Distilled-32B. Specifically, we compare the LLM's success rate at generation (*e.g.*, solving the problem from scratch)

versus its success rate at verification (*e.g.*, writing a verifier that correctly identifies correct and incorrect solutions). The results, together with a manual audit of the filtered properties, are presented in Table 7. Our results empirically confirm this asymmetry of verification. The accuracy of verification (87.0%) is significantly higher than that of code generation (63.0%). Especially on hard problems, the verification accuracy (76.5%) is more than twice the generation accuracy (32.4%). Notably, when applying property validation mentioned in Section 4.2 (w/ filtering), it boosts the verification accuracy on Hard problems from 76.5% to 88.2%, further widening the generation-to-verification gap. The manual audit shows that 88.0% of the filtered properties are correct overall, confirming that filtering resolves many property-generation errors. However, the gap between filtered VerAcc. and manual audit on Medium and Hard problems also indicates that public-test filtering is a practical reliability check rather than a complete soundness guarantee.

*Table 7.* **Asymmetry of Verification.** We evaluate DeepSeek-R1-Distilled-32B's generation and verification accuracy on 100-problem subset of LiveCodeBench. Here, "GenAcc." denotes the standard pass@1 rate of generating the correct solution code. "VerAcc." denotes the accuracy of the model in generating a valid property that correctly distinguishes correct from incorrect solutions. "w/ filtering" denotes using public test cases to filter out invalid properties, and "Manual Audit" denotes human inspection of the filtered properties.

| Difficulty | GenAcc. | VerAcc. | VerAcc. (w/ filtering) | VerAcc. (Manual Audit) |
|---|---|---|---|---|
| Easy ($n = 32$) | 90.6% (29/32) | 93.8% (30/32) | 96.9% (31/32) | 96.9% (31/32) |
| Medium ($n = 34$) | 67.6% (23/34) | 91.2% (31/34) | 94.1% (32/34) | 85.3% (29/34) |
| Hard ($n = 34$) | 32.4% (11/34) | 76.5% (26/34) | 88.2% (30/34) | 82.4% (28/34) |
| **Overall** ($N = 100$) | **63.0%** (63/100) | **87.0%** (87/100) | **93.0%** (93/100) | **88.0%** (88/100) |

**Impact of Input Source.** We further examine whether the gains from input synthesis depend on the same model that performs code generation. During input synthesis, the model only receives the problem description and does not observe the candidate code, while the resulting inputs are evaluated through executable property checks rather than LLM self-evaluation. As shown in Table 8, replacing same-model input synthesis with cross-model input synthesis yields nearly identical performance (76.5% vs. 76.7%). This suggests that PGS's improvement primarily comes from executable property verification over synthesized inputs, rather than from model-specific self-bias in input generation. Using hidden test inputs without labels or outputs provides only a small additional upper bound (77.3%), indicating that the synthesized inputs already recover most of the practical benefit.

*Table 8.* **Ablation study on input synthesis sources.** We evaluate LiveCodeBench with DeepSeek-R1-Distilled-32B. Hidden test inputs are used only as unlabeled inputs without expected outputs.

| Configuration | Input Source | pass@1 (%) |
|---|---|---|
| Direct Generate Baseline | N/A | 64.4 |
| No Input Synthesis | Public tests only | 75.2 |
| Standard Input Synthesis | Same model | 76.5 |
| Cross-Model Input Synthesis | Different LLM | **76.7** |
| Ground-Truth Inputs | Hidden inputs without labels | 77.3 |

## D.2. Multi-Lingual Code Generation

To validate the generalizability of PGS beyond Python, We evaluated PGS on C++ versions of HumanEval (HumanEval-X (Zheng et al., 2023)) and LiveCodeBench, maintaining identical problem logic to their Python counterparts. We adopt 5 refinement iterations, 0.7 temperature and minimal token count strategy for each run. As shown in Table 9, PGS demonstrates consistent improvements across all models and benchmarks. On HumanEval-C/C++ with DeepSeek-R1-Distilled-32B, PGS achieves 96.8% pass@1, outperforming the baseline (92.1%) by 4.7 points. On the more challenging LCB-C/C++, PGS shows even larger gains (PGS 74.8% vs. baseline 62.5%). The fix rate remains robust, averaging 51.4% across C/C++ benchmarks. Thus, PGS maintains its effectiveness across C++ languages, validating its generalizability to diverse code generation scenarios.

*Table 9.* **Multi-Lingual Evaluation on HumanEval-X and LiveCodeBench.** We test PGS's code refinement capability on C/C++ language. All results are reported using mean ± std from 5 runs.

| Method | DeepSeek-Coder-V2 | | Qwen2.5-Coder | | DeepSeek-R1-Distilled-32B | |
| | HumanEval-C | LCB-C | HumanEval-C | LCB-C | HumanEval-C | LCB-C |
|---|---|---|---|---|---|---|
| Baseline | $75.1 \pm 0.8$ | $25.2 \pm 0.9$ | $86.5 \pm 0.7$ | $30.1 \pm 1.0$ | $92.1 \pm 0.6$ | $62.5 \pm 1.0$ |
| CoT | $75.5 \pm 0.6$ | $25.5 \pm 0.8$ | $86.5 \pm 0.6$ | $30.5 \pm 0.9$ | $92.1 \pm 0.5$ | $62.5 \pm 0.9$ |
| Code-T | $79.5 \pm 1.5$ | $27.8 \pm 1.2$ | $87.2 \pm 1.4$ | $32.8 \pm 1.3$ | $93.5 \pm 1.3$ | $68.8 \pm 1.4$ |
| Self-Edit | $80.2 \pm 1.6$ | $28.5 \pm 1.5$ | $89.1 \pm 1.6$ | $33.5 \pm 1.6$ | $94.2 \pm 1.5$ | $71.5 \pm 1.7$ |
| Self-Debugging | $82.5 \pm 1.8$ | $29.8 \pm 1.7$ | $91.5 \pm 1.8$ | $36.8 \pm 1.8$ | $95.5 \pm 1.7$ | $70.8 \pm 1.9$ |
| **PGS (Ours)** | **$87.8 \pm 1.6$** | **$32.8 \pm 1.8$** | **$93.5 \pm 1.6$** | **$38.5 \pm 2.0$** | **$96.8 \pm 1.5$** | **$74.8 \pm 1.9$** |

## D.3. Model Independence and Robustness Analysis

We further investigate the independence of the PGS framework by evaluating its performance across heterogeneous model configurations. This analysis addresses whether the observed gains are contingent on shared underlying biases between the Generator and Tester, or a high dependency on a specific model's capabilities. By using different model families for each role, we examine the framework's robustness to model-pair variations.

The results in Table 10 demonstrate that PGS maintains consistent performance improvements even in cross-model settings. This indicates that the synthesized properties provide sufficiently independent signals to resolve logical flaws, regardless of the architectural similarities between the two agents. Notably, we find that even when a relatively weaker model acts as the Tester (e.g., using Qwen2.5 to provide feedback for DeepSeek-R1-32B), the framework still yields meaningful property-based guidance that improves the pass rate. This strengthens our claim that PGS effectively circumvents the test oracle problem and mitigates the cycle of self-deception. By shifting from raw I/O matching to semantic property validation, the framework ensures a reliable, actionable feedback signal that is not contingent on using a superior model as the judge, thus facilitating more robust code refinement.

*Table 10.* PGS Performance across Heterogeneous Generator-Tester Configurations. The baseline represents the direct inference performance of the Generator without the PGS framework. All reported values are averaged over five independent runs.

| Generator | Tester | HumanEval | MBPP | LiveCodeBench | CodeContests |
|---|---|---|---|---|---|
| Baseline(DS-R1-32B) | - | 93.3 | 73.8 | 64.4 | 38.1 |
| DS-R1-32B | DS-R1-32B | 97.6 | 87.2 | 76.5 | 49.7 |
| DS-R1-32B | Qwen2.5 | 96.8 | 86.6 | 74.9 | 48.4 |
| DS-R1-32B | Qwen3-30B | 97.2 | 87.2 | 76.1 | 49.6 |
| Baseline(Qwen2.5) | - | 87.8 | 65.4 | 31.8 | 12.5 |
| Qwen2.5 | Qwen2.5 | 94.5 | 76.6 | 40.0 | 22.4 |
| Qwen2.5 | DS-V2 | 93.3 | 74.9 | 38.7 | 20.0 |
| Qwen2.5 | DS-R1-32B | 95.9 | 79.8 | 42.7 | 25.3 |

## D.4. More Cost-Effectiveness Analysis

In this section, we provide additional cost-effectiveness results across LiveCodeBench, MBPP, and CodeContests to complement the analysis in the main paper. These benchmarks span different levels of difficulty, allowing us to examine whether PGS maintains a favorable accuracy-cost trade-off beyond a single dataset.

Figure 7 first illustrates the cost-performance trajectory on LiveCodeBench, where PGS reaches higher accuracy with fewer refinement rounds than the baselines. Table 11 further extends this analysis to LiveCodeBench, MBPP, and CodeContests by reporting pass@1 together with the average token cost per problem. On LiveCodeBench, PGS reaches 62.9% pass@1 by the third iteration with 17.4k tokens, already surpassing Self-Edit after five iterations (60.2%, 16.1k tokens) and Self-Debugging after five iterations (61.5%, 39.8k tokens). The pattern is consistent on MBPP and CodeContests: PGS converges to higher accuracy with substantially fewer tokens than Self-Debugging, and it outperforms Self-Edit under comparable cumulative

budgets. For example, on CodeContests, PGS at iteration 3 achieves 39.6% pass@1 with 31.9k tokens, exceeding both Self-Edit at iteration 5 (34.4%, 32.0k tokens) and Self-Debugging at iteration 5 (36.1%, 63.4k tokens). These results indicate that the additional overhead from property generation is offset by faster convergence, yielding a more favorable accuracy-cost trade-off across benchmarks of varying difficulty.

*Table 11.* **Cost-effectiveness analysis across multiple benchmarks.** We report pass@1 and average token cost per problem for **Qwen3-30B** over different refinement budgets. Each cell is formatted as Acc. / Cost.

| Dataset | Method | Iteration 1 | Iteration 3 | Iteration 5 |
|---|---|---|---|---|
| LiveCodeBench | Self-Edit | 56.0% / 6.2k | 58.5% / 11.3k | 60.2% / 16.1k |
|  | Self-Debugging | 57.9% / 11.6k | 60.4% / 25.5k | 61.5% / 39.8k |
|  | PGS (Ours) | **59.3%** / 9.6k | **62.9%** / 17.4k | **65.1%** / 24.2k |
| MBPP | Self-Edit | 75.7% / 3.5k | 77.1% / 7.6k | 78.2% / 11.8k |
|  | Self-Debugging | 76.2% / 9.8k | 78.9% / 19.1k | 80.0% / 30.7k |
|  | PGS (Ours) | **77.8%** / 6.2k | **81.1%** / 12.3k | **82.5%** / 17.5k |
| CodeContests | Self-Edit | 32.5% / 10.7k | 33.8% / 21.5k | 34.4% / 32.0k |
|  | Self-Debugging | 33.2% / 19.4k | 35.7% / 42.6k | 36.1% / 63.4k |
|  | PGS (Ours) | **35.1%** / 16.9k | **39.6%** / 31.9k | **41.7%** / 46.8k |

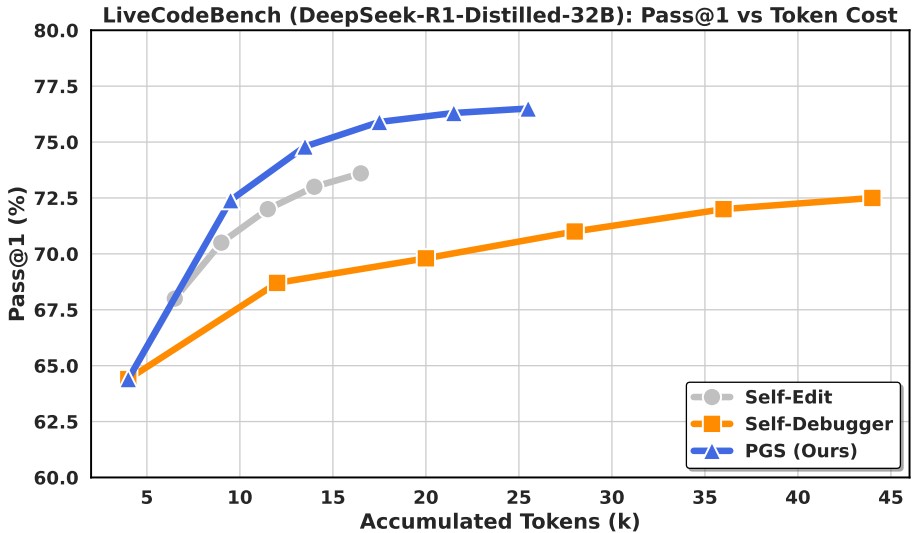

*Figure 7.* **pass@1 performance *vs.* Token Consumption on DeepSeek-R1-32B.** Token counts are calculated using each model's respective tokenizer.

*Table 6.* Comparison on Code Generation across Multiple Benchmarks. We report pass@1 scores with standard deviations. "DS" denotes DeepSeek, and "Claude-4" denotes Claude-Sonnet-4. Fix Rate is calculated as the percentage of problems solved by PGS relative to the failure cases of the Baseline. Empty cells (−) indicate results omitted due to prohibitive cost or incompatibility. The best result in each row is highlighted in **bold**. Results marked with † are cited from original papers; others are reproduced in our experiments.

| Method | DS-V2 | Qwen2.5 | DS-R1-32B | Qwen3-30B | DS-V3.1 | Claude-4 |
|---|---|---|---|---|---|---|
| ***HumanEval (HE)*** | | | | | | |
| Baseline | 76.2 ± 0.7 | 87.8 ± 0.6 | 93.3 ± 0.5 | 91.5 ± 0.8 | 95.5 ± 0.7 | 97.2 ± 0.6 |
| CoT | 76.8 ± 0.5 | 87.8 ± 0.5 | 93.3 ± 0.4 | 91.5 ± 0.7 | 95.6 ± 0.6 | 97.4 ± 0.5 |
| Code-T | 81.1 ± 1.7 | 88.4 ± 1.5 | 94.5 ± 1.4 | 92.2 ± 0.6 | 96.2 ± 0.5 | 97.8 ± 0.4 |
| LDB† | 82.3 | - | - | - | - | - |
| Self-Edit | 81.7 ± 1.6 | 90.2 ± 1.7 | 95.1 ± 1.6 | 92.8 ± 0.7 | 96.5 ± 0.6 | 97.9 ± 0.5 |
| MGDebugger | 83.5 ± 1.7 | 92.1 ± 1.6 | 95.7 ± 1.5 | 93.2 ± 0.8 | 96.8 ± 0.7 | 98.2 ± 0.6 |
| Self-Debugging | 84.1 ± 1.9 | 92.7 ± 1.8 | 96.3 ± 1.7 | 93.5 ± 0.9 | 97.1 ± 0.8 | 98.5 ± 0.7 |
| Reflexion | 86.6 ± 1.2 | 91.5 ± 1.4 | 95.1 ± 1.3 | 92.9 ± 0.6 | 96.9 ± 0.8 | 98.5 ± 0.6 |
| **PGS (Ours)** | **89.0 ± 1.5** | **94.5 ± 1.1** | **97.6 ± 1.0** | **95.2 ± 0.8** | **98.2 ± 0.6** | **99.1 ± 0.3** |
| *Fix Rate* | *53.8%* | *54.9%* | *64.2%* | *43.5%* | *60.0%* | *67.9%* |
| ***MBPP*** | | | | | | |
| Baseline | 56.8 ± 0.6 | 65.4 ± 0.7 | 73.8 ± 0.6 | 73.1 ± 0.9 | 89.5 ± 0.8 | 93.5 ± 0.3 |
| CoT | 57.2 ± 0.4 | 66.6 ± 0.5 | 73.8 ± 0.5 | 73.1 ± 0.8 | 89.8 ± 0.7 | 93.8 ± 0.3 |
| Code-T | 60.4 ± 1.5 | 69.4 ± 1.6 | 82.4 ± 1.0 | 76.5 ± 1.2 | 90.8 ± 1.1 | 94.5 ± 0.5 |
| LDB† | 62.6 | - | - | - | - | - |
| Self-Edit | 62.4 ± 1.5 | 70.2 ± 1.8 | 83.0 ± 1.2 | 78.2 ± 1.3 | 91.2 ± 1.2 | 94.8 ± 0.6 |
| MGDebugger | 63.8 ± 2.0 | 71.2 ± 1.9 | 83.8 ± 1.3 | 79.6 ± 1.4 | 91.5 ± 1.3 | 95.2 ± 0.7 |
| Self-Debugging | 63.8 ± 1.9 | 72.4 ± 2.0 | 84.4 ± 1.4 | 80.0 ± 1.5 | 91.8 ± 1.4 | 95.5 ± 0.8 |
| **PGS (Ours)** | **67.6 ± 1.8** | **76.6 ± 1.9** | **87.2 ± 1.3** | **82.5 ± 1.5** | **94.1 ± 1.4** | **96.5 ± 0.8** |
| *Fix Rate* | *25.0%* | *32.4%* | *51.1%* | *34.9%* | *43.8%* | *46.2%* |
| ***LiveCodeBench (LCB)*** | | | | | | |
| Baseline | 26.7 ± 0.8 | 31.8 ± 0.9 | 64.4 ± 0.9 | 52.2 ± 1.0 | 72.5 ± 2.6 | 63.1 ± 1.7 |
| CoT | 26.9 ± 0.9 | 32.4 ± 0.8 | 64.4 ± 0.8 | 52.2 ± 0.9 | 72.7 ± 2.2 | 63.2 ± 1.6 |
| Code-T | 29.2 ± 1.3 | 34.6 ± 1.4 | 70.8 ± 1.6 | 54.5 ± 1.8 | 75.5 ± 2.7 | 68.2 ± 1.4 |
| Self-Edit | 30.2 ± 1.9 | 35.2 ± 1.8 | 73.6 ± 1.8 | 60.2 ± 2.0 | 79.8 ± 2.8 | 70.5 ± 1.9 |
| Self-Debugging | 31.3 ± 2.2 | 38.5 ± 2.0 | 72.5 ± 2.0 | 61.5 ± 2.1 | 80.5 ± 2.0 | 72.8 ± 2.0 |
| **PGS (Ours)** | **34.1 ± 1.7** | **40.0 ± 1.9** | **76.5 ± 1.8** | **65.1 ± 1.4** | **83.2 ± 2.0** | **75.5 ± 1.8** |
| *Fix Rate* | *10.1%* | *12.0%* | *34.0%* | *27.0%* | *38.9%* | *33.6%* |
| ***CodeContests (CC)*** | | | | | | |
| Baseline | 12.5 ± 0.6 | 14.4 ± 0.7 | 38.1 ± 0.8 | 30.8 ± 0.9 | 46.8 ± 1.1 | 42.1 ± 1.0 |
| CoT | 12.8 ± 0.8 | 14.9 ± 0.8 | 38.1 ± 0.8 | 30.8 ± 0.7 | 47.2 ± 1.0 | 42.6 ± 0.9 |
| Code-T | 14.2 ± 0.9 | 15.9 ± 1.0 | 42.9 ± 1.4 | 33.2 ± 1.6 | 49.6 ± 1.4 | 44.3 ± 1.5 |
| Self-Edit | 15.6 ± 1.7 | 16.4 ± 1.5 | 44.8 ± 1.6 | 34.4 ± 1.7 | 53.2 ± 1.6 | 48.5 ± 1.6 |
| Self-Debugging | 16.1 ± 2.1 | 17.3 ± 1.9 | 45.8 ± 1.9 | 36.1 ± 2.0 | 54.5 ± 1.8 | 49.9 ± 1.9 |
| **PGS (Ours)** | **20.2 ± 1.8** | **22.4 ± 2.0** | **49.7 ± 2.0** | **41.7 ± 2.2** | **60.2 ± 2.1** | **55.9 ± 2.1** |
| *Fix Rate* | *8.8%* | *9.3%* | *18.7%* | *15.8%* | *25.2%* | *23.8%* |
| ***SWE-bench*** | | | | | | |
| SWE-agent | 9.8 ± 1.4 | 10.3 ± 0.8 | 34.4 ± 2.0 | 46.5 ± 1.7 | 54.2 ± 1.8 | 65.5 ± 1.6 |
| **PGS (Ours)** | **11.9 ± 1.0** | **12.8 ± 0.5** | **37.3 ± 2.3** | **50.7 ± 1.9** | **58.4 ± 2.5** | **70.2 ± 1.5** |

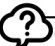

Question:
You are given a 0-indexed binary string s of length n on which you can apply two types of operations:
Choose an index i and invert all characters from index 0 to index i (both inclusive), with a cost of i + 1
Choose an index i and invert all characters from index i to index n - 1 (both inclusive), with a cost of n - i
Return the minimum cost to make all characters of the string equal.
Invert a character means if its value is '0' it becomes '1' and vice-versa.

```
class Solution:
    def minimumCost(self, s: str) -> int:
        n = len(s)
        transitions = []
        for i in range(n - 1):
            if s[i] != s[i + 1]:
                transitions.append(i)

        if not transitions:
            return 0

        cost0 = 0  # cost to make current segment 0
        cost1 = float('inf')  # cost to make current segment 1

        for i in transitions:
            prefix_cost = i + 1
            suffix_cost = n - i

            # Calculate new costs based on flipping prefix or suffix
            new_cost0 = min(cost1 + prefix_cost, cost1 + suffix_cost)
            new_cost1 = min(cost0 + prefix_cost, cost0 + suffix_cost)

            cost0, cost1 = new_cost0, new_cost1

        # Check the final segment after the last transition
        last_i = transitions[-1]
        final_length = n - (last_i + 1)
        if final_length > 0:
            final_prefix = last_i + 2
            final_suffix = n - (last_i + 1)
            final_char = s[last_i + 1]
            if final_char == '0':
                # For cost1, need to flip the final segment
                cost1 += min(final_prefix, final_suffix)
            else:
                # For cost0, need to flip the final segment
                cost0 += min(final_prefix, final_suffix)

        result = min(cost0, cost1)

        return result
```

## Code with Properties

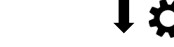

```
class Solution:
    def minimumCost(self, s: str) -> int:
        n = len(s)
        transitions = []
        for i in range(n - 1):
            if s[i] != s[i + 1]:
                transitions.append(i)

        if not transitions:
            return 0

        cost0 = 0  # cost to make current segment 0
        cost1 = float('inf')  # cost to make current segment 1

        for i in transitions:
            prefix_cost = i + 1
            suffix_cost = n - i

            # Calculate new costs based on flipping prefix or suffix
            new_cost0 = min(cost1 + prefix_cost, cost1 + suffix_cost)
            new_cost1 = min(cost0 + prefix_cost, cost0 + suffix_cost)

            cost0, cost1 = new_cost0, new_cost1

        ......(13 lines left)......

        result = min(cost0, cost1)

        # Assertion checks to ensure correctness
        assert result >= 0, f"Negative cost detected: {result}"
        if not transitions:
            assert result == 0, f"Uniform string, expected cost 0, got
{result}"
        else:
            assert result > 0, "Result should be positive when
transitions exist"
        assert not math.isinf(cost0), "cost0 must be finite"
        assert not math.isinf(cost1), "cost1 must be finite"
        return result
```

## Correct Code

```
class Solution:
    def minimumCost(self, s: str) -> int:
        n = len(s)
        total_cost = 0

        # Iterate through the string to find all transition points.
        for i in range(n - 1):
            if s[i] != s[i+1]:
                # A transition exists at index 'i'.
                # Cost to fix it by flipping the prefix s[0...i].
                prefix_flip_cost = i + 1

                # Cost to fix it by flipping the suffix s[i+1...n-1].
                # This is operation 2 starting at index i+1.
                suffix_flip_cost = n - (i + 1)

                # Add the minimum of the two costs to the total.
                total_cost += min(prefix_flip_cost, suffix_flip_cost)

        return total_cost
```

## Results of Execution

```
{
    "error": "AssertionError('cost0 must be finite')",
    "error_message": "Runtime Error",
    "inputs": "['0011']",
    "expected": "2"
}
```

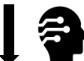

## Model Output

```
<think>
We are given a binary string `s`. We can perform two operations:
...(3292 tokens left)...
</think>
```

*Figure 8.* **A Case Study on the PGS Workflow.** This figure illustrates the end-to-end refinement process for a challenging problem. **(1)** The process starts with an initial, complex, and incorrect solution. **(2)** This code is then instrumented with property-based assertions. **(3)** Executing the instrumented code with a minimal input triggers a clear `AssertionError`, transforming a latent logic bug into an explicit failure signal. **(4)** This targeted feedback enables the LLM to effectively debug the program, ultimately producing a **(5)** much simpler and correct final solution.

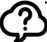

Question:
You are given a 0-indexed integer array receiver of length n and an integer k.
There are n players having a unique id in the range [0, n - 1] who will play a ball passing game, and receiver[i] is the id of the player who receives passes from the player with id i. Players can pass to themselves, i.e. receiver[i] may be equal to i. You must choose one of the n players as the starting player for the game, and the ball will be passed exactly k times starting from the chosen player.
For a chosen starting player having id x, we define a function f(x) that denotes the sum of x and the ids of all players who receive the ball during the k passes, including repetitions.
In other words, f(x) = x + receiver[x] + receiver[receiver[x]] + ... + receiver^(k)[x].

```
class Solution:
    def getMaxFunctionValue(self, receiver: list[int], k: int) -> int:
        n = len(receiver)
        visited = [False] * n
        p = [0] * n
        sum_p = [0] * n
        m = [0] * n
        sum_c = [0] * n
        cycle_prefix_sums = [[] for _ in range(n)]
        cycle_index = [-1] * n
        for x in range(n):
            if not visited[x]:
                path = []
                current = x
                while True:
                    if visited[current]:
                        if current in path:
                            # Precompute prefix sums for the cycle
                            ......(28 lines left)......
                        else:
                            # Process the tree path leading to a visited node
                            ......(11 lines left)......
                    else:
                        visited[current] = True
                        path.append(current)
                        current = receiver[current]
        max_f = 0
        best_x = -1
        for x in range(n):
            total_steps = k + 1
            if total_steps <= p[x]:
                current_sum = sum_p[x]
            else:
                pre_steps = p[x]
                full_cycles = (total_steps - pre_steps) // m[x]
                remaining_steps = (total_steps - pre_steps) % m[x]
                current_sum = sum_p[x] + full_cycles * sum_c[x]
                if remaining_steps > 0:
                    current_sum +=
cycle_prefix_sums[x][remaining_steps]
            if current_sum > max_f:
                max_f = current_sum
                best_x = x
        return max_f
```

## Code with Properties

```
class Solution:
    def getMaxFunctionValue(self, receiver: list[int], k: int) -> int:
        ......(57 lines left)......
        max_f = 0
        best_x = -1
        for x in range(n):
            total_steps = k + 1
            if total_steps <= p[x]:
                current_sum = sum_p[x]
            else:
                pre_steps = p[x]
                full_cycles = (total_steps - pre_steps) // m[x]
                remaining_steps = (total_steps - pre_steps) % m[x]
                current_sum = sum_p[x] + full_cycles * sum_c[x]
                if remaining_steps > 0:
                    current_sum +=
cycle_prefix_sums[x][remaining_steps]

            # Property Verification 1: Sum must be non-negative
            assert current_sum >= 0, f"Negative sum for x={x}"

            # Property Verification 2:
            # For small k, computed sum must match simulation
            if k <= 100:
                def simulate_sum(receiver, x, k):
                    total = x
                    current = x
                    for _ in range(k):
                        current = receiver[current]
                        total += current
                    return total

                simulated_sum = simulate_sum(receiver, x, k)
                assert simulated_sum == current_sum,
                f"Simulated sum {simulated_sum} does not match
computed {current_sum} for x={x}"
            if current_sum > max_f:
                max_f = current_sum
                best_x = x

        return max_f
```

## Correct Code

```
class Solution:
    def getMaxFunctionValue(self, receiver: List[int], k: int) -> int:
        n = len(receiver); m = 60
        dp = [[0] * n for _ in range(m)]
        dp_sum = [[0] * n for _ in range(m)]
        for i in range(n):
            dp[0][i] = receiver[i]
            dp_sum[0][i] = receiver[i]
        for j in range(1, m):
            for i in range(n):
                mid = dp[j-1][i]
                dp[j][i] = dp[j-1][mid]
                dp_sum[j][i] = dp_sum[j-1][i] + dp_sum[j-1][mid]
        ans = -10**25
        for x in range(n):
            cur = x; s = 0
            for j in range(m):
                if (k >> j) & 1:
                    s += dp_sum[j][cur]
                    cur = dp[j][cur]
            candidate = x + s
            if candidate > ans:
                ans = candidate
        return ans
```

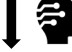

## Results of Execution

```
{
    "error": "AssertionError('Simulated sum 4 does not
match computed 5 for x=1')",
    "error_message": "Runtime Error",
    "inputs": "[[2, 0, 1], 4]",
    "expected": "6"
}
```

## Model Output

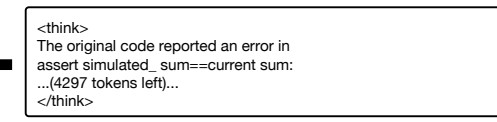

```
<think>
The original code reported an error in
assert simulated_ sum==current sum:
...(4297 tokens left)...
</think>
```

*Figure 9.* **A Case Study on the PGS Workflow.** This figure illustrates the end-to-end refinement process for a challenging problem. **(1)** The process starts with an initial, complex, and incorrect solution. **(2)** This code is then instrumented with property-based assertions. **(3)** Executing the instrumented code with a minimal input triggers a clear `AssertionError`, transforming a latent logic bug into an explicit failure signal. **(4)** This targeted feedback enables the LLM to effectively debug the program, ultimately producing a **(5)** much simpler and correct final solution.

*Table 12*. **Prompt Templates for Code Generation.** This table illustrates how prompts are constructed. Placeholders like {question_content} and {starter_code} are replaced with actual content during runtime.

---

**1. Core User Prompt Template (Generated by `get_generic_question_template_answer`)**

This is the base structure for the user's request, with a key variation based on the presence of starter code.

**### Question:**
{question_content}

**IF** starter_code is provided:
**### Format:** You will use the following starter code to write the solution...
```python\n{starter_code}\n```
**ELSE**:
**### Format:** Read the inputs from stdin solve the problem and write the answer to stdout...
```python\n# YOUR CODE HERE\n```

**### Answer:** (use the provided format with backticks)

- - - - - - - - - - - - - - - - - - - - - - - - - - - - - - - - - - - - - - - - - - - - - - - - - - - - - - - - -

**2. Model-Specific System Messages (From `PromptConstants`)**

These messages instruct the model on its role and expected output format.

**DeepSeek-R1 Style:**
```
<|begin of sentence|>A conversation between User and Assistant...  <think>
reasoning process here </think> <answer> answer here </answer>.<|User|>
```
**CodeQwen Style:**
```
<|im_start|>system\nYouare a helpful assistant.  <|im_end|>\n<|im_start|>user
```

- - - - - - - - - - - - - - - - - - - - - - - - - - - - - - - - - - - - - - - - - - - - - - - - - - - - - - - - -

*Table 13.* **Prompt Templates for Property Generation.** This table illustrates the layered construction of prompts that instruct an LLM to generate Python functions for property validation.

---

**1. Core User Prompt for Property Generation**

This is the base instruction set given to the model, asking it to act as a testing expert.

**### Task Description:**
You are a software testing expert. Your task is to analyze the problem description and generate a Python function that asserts a specific property or invariant a correct solution must satisfy. This property-checking function should take the candidate solution's input and output, returning `True` if the property holds, or `False` if it fails.

**### Inputs Provided for Context:**
- Problem Description: {`question`}
- Example of a Correct Solution's Input/Output: {`example_solution_io`}

------------------------------------------------------------------------

**2. Property Type-Specific Guidance**

The core prompt is refined with examples based on the type of property required.

**Relational Property Example (e.g., for a sorting problem):**
The property checks the relationship between input and output.

```
# Generated code checks if output is a sorted permutation of input
def check_property(input_list, output_list):
    return sorted(input_list) == output_list
```

**Intrinsic Property Example (e.g., for prime factorization):**
The property checks a characteristic of the output itself.

```
# Generated code checks if the product of factors equals the input
def check_property(n, factors):
    product = 1
    for factor in factors:
        product *= factor
    return product == n
```

------------------------------------------------------------------------

**3. Model-Specific System Messages & Wrappers**

Finally, the entire prompt is wrapped with model-specific system messages and formatting tags.

**DeepSeek-R1 Style:**
```
<|begin of sentence|>A conversation...  The assistant first thinks...  <think>
reasoning process here </think> <answer> answer here </answer>.<|User|> [Core
Prompt + Property Guidance] <|Assistant|>
```
**CodeQwen Instruct Style:**
```
<|im_start|>system\nYouare a helpful AI...<|im_end|>
<|im_start|>user\n[Core Prompt + Property Guidance]<|im_end|>
<|im_start|>assistant
```

*Table 14.* **Prompt Templates for Dynamic Input Script Generation.** This table shows the layered construction of prompts that instruct an LLM to generate a Python script, which in turn produces randomized test inputs.

---

**1. Core User Prompt for Input Script Generation**

This is the base instruction set given to the model, outlining the primary task.

**### Task Description:**
You are an expert Python programmer. Your task is to write a Python script that utilizes randomization (seeded by current time) to generate diverse and valid input strings. The script's standard output must be a single string formatted exactly as required.

**### Inputs Provided for Context:**
- Problem Description: {question}
- Original Code Snippet: {original_code_snippet}
- Target Platform: {platform}
- Example Input String: {example_input_str}

- - - - - - - - - - - - - - - - - - - - - - - - - - - - - - - - - - - - - - - - - - - - - - - - - - - - - - - - - -

**2. Platform-Specific Formatting Guidance**

The core prompt is augmented with specific instructions based on the target platform's input format.

**LeetCode / MBPP / HumanEval Style Example:**
Your Python script should generate a string where each line is a JSON object.

```
Example Target Output String:
[1,2,3]
"some_string"
```

**Codeforces / AtCoder Style Example:**
Your Python script should generate a string with space- or newline-separated values.

```
Example Target Output String:
3
1 2 3
```

- - - - - - - - - - - - - - - - - - - - - - - - - - - - - - - - - - - - - - - - - - - - - - - - - - - - - - - - - -

**3. Model-Specific System Messages & Wrappers**

Finally, the entire prompt is wrapped with model-specific system messages and formatting tags.

**DeepSeek-R1 Style:**
```
<|begin of sentence|>A conversation...  The assistant first thinks...  <think>
reasoning process here </think> <answer> answer here </answer>.<|User|> [Core
Prompt + Platform Guidance] <|Assistant|>
```
**CodeQwen Instruct Style:**
```
<|im_start|>system\nYouare a helpful AI...<|im_end|>
<|im_start|>user\n[Core Prompt + Platform Guidance]<|im_end|>
<|im_start|>assistant
```

*Table 15.* **Prompt Templates for Feedback-Driven Code Repair.** This table outlines the prompt structure for guiding an LLM to debug and fix erroneous code based on specific execution feedback.

---

**1. Core Repair Prompt Structure**

This is the main template that presents the problem, the buggy code, and the specific error context to the model.

**### Question:**
```
{question}
```

**### Buggy Code:**
```
```python\n{buggy_code}\n```
```

**### Error Context:**
```
{error_feedback_from_part_2}
```

**### Your Task:**
First, provide a concise explanation of the error. Then, generate the entire corrected program.

- - - - - - - - - - - - - - - - - - - - - - - - - - - - - - - - - - - - - - - - - - - - - - - - - - - - - - - - - - -

**2. Dynamic Error Feedback Generation**

This component translates a structured error object ('metadata') into a human-readable feedback string.

**IF** `error_code == -2 (Wrong Answer)`:
Context: The program previously produced a wrong answer.
Input: {inputs}
Generated Output: {output}
Expected Output: {expected}

**ELSE**:
Context: The program previously encountered a runtime error.
Input: {inputs}
Error Details: {error}

- - - - - - - - - - - - - - - - - - - - - - - - - - - - - - - - - - - - - - - - - - - - - - - - - - - - - - - - - - -

**3. Model-Specific System Messages & Wrappers**

Finally, the entire prompt is wrapped with model-specific system messages and formatting tags.

**DeepSeek-R1 Style:**
```
<|begin of sentence|>A conversation...  The assistant first thinks...  <think>
reasoning process here </think> <answer> answer here </answer>.<|User|> [Core
Prompt + Platform Guidance] <|Assistant|>
```

**CodeQwen Instruct Style:**
```
<|im_start|>system\nYouare a helpful AI...<|im_end|>
<|im_start|>user\n[Core Prompt + Platform Guidance]<|im_end|>
<|im_start|>assistant
```

