# OpenReview forum: "PGS: Effective LLM Code Refinement via Property-Oriented and Structurally Minimal Feedback"
_ICML.cc/2026/Conference — ICML 2026 regular_

### Official Review · Reviewer_5dwM · 2026-03-06

**Soundness:** 3
**Presentation:** 3
**Significance:** 3
**Originality:** 3
**Overall Recommendation:** 5
**Confidence:** 3

**Summary:**

This work introduces Property-Generated Solver (PGS), a framework for refining LLM-generated code. Instead of providing test execution outcomes, PGS provides two kinds of high-quality feedback: (1) property-oriented feedback that frames failures as violations of high-level semantic rules, and (2) structurally minimal feedback that selects the simplest possible failing counterexample. The implementation is a two-agent system (a Generator that writes code and a Tester that validates properties and formulates feedback) that iterates up to five times per problem. Evaluated across five benchmarks and six LLMs, PGS achieves a bug fix rate 1.4x - 1.6x higher than the best existing debugging approaches, establishing a new state-of-the-art in automated code refinement.

**Compliance With Llm Reviewing Policy:**

Affirmed.

**Final Justification:**

The authors have addressed my concerns during rebuttal and I have adjusted my score accordingly. My final recommendation is Accept.

**Key Questions For Authors:**

See weaknesses.

**Limitations:**

I could not find a limitations section. Can you please point me to it in your paper, if any?

**Strengths And Weaknesses:**

Strengths:
- The idea of PGS is novel
- Evaluation is rigorous and covers multiple benchmarks
- Pilot Study in the paper is valuable

Weaknesses:
- Why should someone trust using "a powerful LLM for test input generation" in Input Synthesis for Property Probing? Any study on how good these inputs are?
- What is the cost of running your tool in terms of input/output token consumption, runtime, $ cost, etc? I am not sure about how practical PGS is.
- The delta of improvement seems very small. How would you convince someone its not?

---

> ### Author Rebuttal · Authors · 2026-03-31
>
> ### **Q1: Trustworthiness and Quality of "Input Synthesis"**
>
> - To clarify, the "powerful LLM" used for input synthesis in PGS is the Generator. We will refine this phrasing in the revision to avoid ambiguity.
>
> - Leveraging the model's own reasoning to synthesize inputs is a widely adopted practice in recent literature [1, 2]. Our results demonstrate that these inputs, when coupled with property-oriented feedback, are highly effective at exposing logical flaws.
>
> [1] OpenAI. "Competitive Programming with Large Reasoning Models."  arXiv preprint arXiv:2502.06807
>
> [2] Liu, Kaibo, et al. "Llm-powered test case generation for detecting tricky bugs." arXiv preprint arXiv:2404.10304.
>
> ---
>
> ### **Q2: Cost Analysis and Practicality**
>
> We provide a cost analysis in **Figure 5 and Appendix D.5 (Figure 7)**. Although PGS has higher per-round token consumption than simple baselines (e.g., Self-Edit), its convergence is substantially faster. For example,
>
> - On HumanEval, PGS surpasses Self-Edit's 5-round performance with only 2 iterations and lower total token cost.
>
> - On LiveCodeBench, PGS achieves with ~25k tokens what Self-Debugger fails to reach even with 40k+ tokens, thereby reducing overall computational expenditure.
>
> This demonstrates PGS's practicality for real-world deployment.
>
> ---
>
> ### **Q3: Significance of Improvements**
>
> The performance gains primarily stem from two factors: the base model's capability and the effectiveness of refinement method.
>
> - **When Using Weaker Base Models (e.g. DeepSeek-V2)**
>
>   - On simpler datasets such as HumanEval, PGS achieves an absolute improvement of 3-4% over competing approaches.
>
>   - On more challenging benchmarks like CodeContest, the relative gains are substantially more pronounced. The number of successfully repaired samples effectively doubles (e.g., from 12.5%→16.1% to 12.5%→20.2%).
>
> - **When Using Stronger Base Models  (e.g. Claude-4)**
>
>   - The performance on simple benchmarks approaches saturation (near or exceeding 95% pass rates), which makes the comparisons less meaningful. Nevertheless, PGS continues to demonstrate the best performance among all baselines.
>
>   - On challenging benchmarks, PGS further solidifies its advantage (e.g., from 42.1%→49.9% to 42.1%→55.9% in CodeContest), exhibiting a trend where the difficulty of the task correlates positively with the magnitude of improvement.
>
> Overall, PGS demonstrates robustness across the spectrum of base model capabilities and task difficulties, with its advantages becoming increasingly pronounced as the complexity of the debugging challenge grows.
>
> ---
>
> ### **Q4: Missing Limitation Section**
>
> We will add the limitation section in the revised version to discuss:
>
> - **Computation Cost.** PGS introduces additional overhead due to property generation and input synthesis. However, as demonstrated in our cost-effectiveness analysis, the performance gains provides a favorable cost-benefit ratio compared to baseline methods.
>
> - **Property Correctness.** The accuracy of generated properties can be affected by the underlying LLM's reasoning capabilities. To address this, we currently employ a filtering strategy using public test cases. We acknowledge this remains an area for improvement and will explore more robust verification mechanisms in future work.

---

> > ### Author Rebuttal · Reviewer_5dwM · 2026-04-01
> >
> > Thank you for addressing my concerns. I believe my questions Q2, Q3 and Q4 are addressed. However, I am skeptical on "using the model's reasoning to synthesize inputs". My main concern is LLMs are biased towards their own reasoning [1,2], and they tend to favor their decision even if incorrect. It would be better if this claim can be justified with an ablation in your study. I stand firm with my original score.
> >
> > [1] Pan, Alexander, Erik Jones, Meena Jagadeesan, and Jacob Steinhardt. "Feedback loops with language models drive in-context reward hacking." arXiv preprint arXiv:2402.06627 (2024).
> >
> > [2] Liu, Yiqi, Nafise Sadat Moosavi, and Chenghua Lin. "LLMs as narcissistic evaluators: When ego inflates evaluation scores." In Findings of the Association for Computational Linguistics: ACL 2024, pp. 12688-12701. 2024.

---

> > > ### Author Response · Authors · 2026-04-03
> > >
> > > ### **Clarification on "Input Synthesis Mechanism"**
> > >
> > > **Pan et al. [1]** and **Liu et al. [2]** demonstrate self-bias in LLM self-evaluation loops where models assess their own outputs. PGS avoids this through architectural isolation and external verification.
> > >
> > > - **Blind Input Synthesis and External Verification.** During our input synthesis, the LLM receives only the **problem description without access to the Generator's code**. Inputs are synthesized based on problem constraints (e.g., "generate a random array of length N") rather than code-specific errors. This fundamentally blocks reward hacking: without access to the code, the LLM cannot tailor inputs to exploit specific failure patterns as described by **Pan et al.** Moreover, our evaluation relies on **executable script execution**, which is fundamentally distinct from the "narcissistic evaluation" (self-scoring) criticized by **Liu et al.**, and is immune to LLM preference biases.
> > >
> > > - **Ablation Study on Replacing LLM-Synthesized Inputs.** To systematically validate the impact of input synthesis and isolate potential self-bias effects, we conducted an ablation study on LiveCodeBench benchmark used by Deepseek-R1-32B model. As shown in **Table D below**, comparing five configurations reveals a negligible performance gap between **(c) Standard Input Synthesis** and **(d) Cross-Model Input Synthesis** ($\sim$0.2%) demonstrates that the choice of input synthesis model does not affect the outcome, confirming that property verification rather than the synthesis model's identity drives performance robustness.
> > >
> > > | Configuration | Input Source | Description | Pass@1 (%) |
> > > |:---|:---|:---|:---|
> > > | (a) Direct Generate Baseline | N/A | Direct code generation without iterative refinement | 64.4% |
> > > | (b) No Input Synthesis | Public tests only | Feedback from public tests only; no LLM synthesis | 75.2% |
> > > | (c) Standard Input Synthesis | Same model | Standard PGS: same model for input synthesis | 76.5% |
> > > | (d) Cross-Model Input Synthesis | Different LLM | Tester uses Qwen3-30B for input synthesis | 76.7% |
> > > | (e) Ground-Truth Inputs | Private test cases | Uses hidden test inputs (no labels/outputs) | 77.3% |
> > >
> > > Table D: Ablation Study on Input Synthesis Sources
> > >
> > > [1] Pan, Alexander, Erik Jones, Meena Jagadeesan, and Jacob Steinhardt. "Feedback loops with language models drive in-context reward hacking." ICML 2024.
> > >
> > > [2] Liu, Yiqi, Nafise Sadat Moosavi, and Chenghua Lin. "LLMs as narcissistic evaluators: When ego inflates evaluation scores." ACL 2024.

---

### Official Review · Reviewer_cNuW · 2026-03-07

**Soundness:** 3
**Presentation:** 3
**Significance:** 2
**Originality:** 2
**Overall Recommendation:** 3
**Confidence:** 4

**Summary:**

The paper argues that the main bottleneck is not the number of generated tests but the quality of the feedback used for repair. It proposes Property-Generated Solver (PGS), a two-agent framework in which a Tester derives high-level properties from the problem statement, validates them against public tests, synthesizes probing inputs, and returns the smallest failing counterexample as feedback to a Generator that revises the code. The paper supports this design with a pilot study showing that property-oriented feedback is more effective than simple input-output feedback, and that choosing the minimal counterexample by input token count works better than other minimization proxies. It then evaluates PGS across five benchmarks and multiple models, reporting consistent gains over direct prompting, TDD-style refinement, and debugger-style self-correction baselines.

**Compliance With Llm Reviewing Policy:**

Affirmed.

**Key Questions For Authors:**

How often are the generated properties actually correct after public-test filtering? A manual audit, even on a subset, would help a lot here.

How much of the gain comes from the form of the feedback itself, and how much comes from giving the tester side a more elaborate role and more computation?

What do the generated properties look like on SWE-bench in practice? I would like to understand how this behaves in repository-level settings, not just function-level problems.

What are the main failure modes of PGS? In particular, are there cases where wrong properties or over-minimized counterexamples actively hurt repair?

I would encourage the authors to position the contribution more carefully relative to older work on property-based testing, invariant inference, counterexample minimization, and specification-guided repair. Why should this be viewed as more than a recombination of those ideas?

**Limitations:**

The current discussion is too thin. The paper should say more about incorrect properties, weak public-test filtering, extra cost, overfitting to generated checks, and misplaced trust in repaired code.

**Strengths And Weaknesses:**

I like the paper’s core intuition. The idea that repair quality depends heavily on the form of the feedback, not just on whether some test failed, is sensible and worth studying.  The pilot study is useful: it asks whether property-oriented feedback helps more than ordinary I/O feedback, and whether smaller counterexamples are indeed better than larger ones. The broader experimental section is also fairly comprehensive by current ML-for-code standards, and the reported gains look meaningful.

My main reservation is that the paper overstates how new this is, especially if judged against software engineering or PL work rather than recent LLM-for-code papers. The exact end-to-end assembly may be new, but the underlying ingredients are not. The method is built from ideas that have been around for a long time: property-based testing, inferred invariants/specifications, shrinking or minimizing failing counterexamples, and semantics- or specification-guided repair. In that sense, the contribution felt to me more like a well-executed LLM-era recombination than a genuinely new repair paradigm.
https://dl.acm.org/doi/epdf/10.1145/351240.351266

https://dl.acm.org/doi/epdf/10.1145/3660791

https://arxiv.org/pdf/2410.13542

https://arxiv.org/pdf/2502.18917

The main weakness is that the method leans heavily on LLM-generated properties, but the validation of those properties is fairly weak. Filtering them against public tests is a reasonable step, but it does not tell us how often the surviving properties are incomplete, slightly wrong, or overly restrictive while still remaining consistent with the visible tests. Since the whole method depends on those properties being good enough to act as a semantic guide, I wanted a much clearer audit here.

I had a similar reaction to some of the stronger wording in the paper. Claims about “breaking the cycle of self-deception” feel stronger than the evidence. The method is still an all-LLM loop. The heterogeneous Generator/Tester setup is a useful check, but it does not really settle the causal story. Some of the gains may come from the structure of the feedback, but some may also come from the fact that the tester side is doing more work and consuming more budget. I think the paper would be stronger if it made a more modest claim here.

---

> ### Author Rebuttal · Authors · 2026-03-31
>
> ### **Q1: Clarification on PGS's Novelty**
> - We acknowledge that PGS draws inspiration from classic software engineering techniques (property-based testing, delta-debugging), which we have properly cited. However, our core contribution is not a simple recombination, but **constructing an actionable feedback system through property-based testing for code refinement**.
> - To the best of our knowledge, PGS is the first work to apply property-based testing in LLM-based code refinement. Previous TDD methods in code agents use concrete I/O test cases (oracle prediction) to formulate feedback, but they often suffer from sparse or noisy signals when high-quality test cases are unavailable. In contrast, we shift the focus to property-based verification, which provides a more robust and semantically rich verification without oracle prediction by defining the invariant logic of the program.
> - Besides, through pilot study and extensive experiments, PGS validates the feasibility of property-based testing and delta debug principle in LLM-based code refinement. These findings constitute a methodological foundation for feedback system design targeting autonomous code agents, rather than merely an LLM-era recombination of classic techniques.
>
> ---
> ### **Q2: Claims about "breaking self-deception"**
> We will rephrase the "breaking self-deception" to "mitigating self-deception" in revised paper. Thanks for your advice.
>
> ---
> ### **Q3: Property Validity and Quality**
> - We evaluated property accuracy in **Appendix D.1 (Table 6)** on 100 LiveCodeBench cases, showing the verification accuracy (VerAcc) before and after public-test filtering. The VerAcc reaches 93.0% after filtering.
> - Moreover, we perform a manual audit (**Table C** below) to check the correctness of the generated properties in above 100 LiveCodeBench cases. The results show that our filtering mechanism effectively resolves >80% of the issues in property generation. This suggests that the current performance bottleneck lies in the model's inherent reasoning capabilities rather than the accuracy of the properties themselves (see **Q6** for a detailed discussion). We acknowledge that property correctness is a key issue, and we will study more advanced verification in future work.
>
> |Difficulty|GenAcc.|VerAcc. (after filtering)|VerAcc. (Manual Audit)|
> |:---|:---|:---|:---|
> |Easy|90.6%|96.9%|96.9%|
> |Medium|67.6%|94.1%|85.3%|
> |Hard|32.4%|88.2%|82.4%|
> |Overall|63.0%|93.0%|88.0%|
>
> Table C: Extended Analysis on Verification
>
> ---
> ### **Q4:  Source of Performance Gain**
> To quantify the proportion of performance gains (from proposed feedback or computation), we can adopt Self-Edit as the raw I/O-based feedback baseline, and further increases its budget (more refinement iterations) to compare with the results of PGS.
> - As shown in **Appendix Figure 7**, PGS at Iteration 2 (74.8% Acc / 13.3k Tokens Cost) outperforms Self-Edit at Iteration 5 (73.6% Acc / 16.8k Tokens Cost) in terms of both accuracy and efficiency.
> - We observe that simply increasing the debug budget for I/O-based testing provides limited performance gains, and its effectiveness significantly lags behind that of property-oriented feedback. This suggests that the majority of our performance gains stem from the high-quality property feedback itself rather than increased computational cost.
>
> ---
> ### **Q5: Property in SWE-bench**
> In SWE-bench, generated properties evolve from math constraints to behavioral specifications or API contracts extracted from the issue description. For instance, in a Django project, a property might be "a specific method must raise a ValidationError on invalid input". These properties guide the Tester in generating complex probing inputs and minimal counterexamples, providing the Generator with more semantically rich repair guidance. We will add examples to the revised paper.
>
> ---
> ### **Q6: Failure mode analysis**
> - **Failure Modes.** We identify two primary failure modes:
>   - **Property Coverage.** Due to limitations in model reasoning capabilities, the generated properties often struggle to achieve full coverage of all potential bugs in complex tasks, particularly for edge cases.
>   - **Generator Stagnation.** In difficult problems, even with correct properties/counterexamples, the Generator fails to code the logic, often due to highly complex program structures or specialized algorithmic logic.
> - **Wrong Properties or Over-Minimized Counterexamples Hurt Repair?**
>   - **Wrong Properties.** We rarely observe cases where incorrect properties hindered the repair process. Even if flawed properties are generated, their negative impact is largely mitigated through public test case filtering and iterative refinement.
>   - **Over-Minimized Counterexamples.** Across all models and baselines, "minimal counterexamples" solved the most problems compared to other strategies. Thus, repair failures primarily stem from the inherent limits of the model's capability rather than the minimization feedback itself.

---

> > ### Author Rebuttal · Reviewer_cNuW · 2026-04-04
> >
> > The rebuttal successfully answers several concerns however I'm still not convinced about conceptual novelty and the sufficiency of property validation in complex settings so I will keep my score.

---

> > > ### Author Response · Authors · 2026-04-07
> > >
> > > ### **Q7: Conceptual Novelty**
> > >
> > > PGS is not a mere recombination of classic SE techniques with LLMs, but the first framework that leverages PBT specifically to address the "cycle of self-deception" in LLM code refinement (see **Table E** for detailed comparisons with the four works you mentioned). The key innovation is an **asymmetric architecture that extracts logical invariants from problem descriptions to break LLM logical hallucinations**, specifically targeting the LLM-specific phenomenon of models being confidently wrong. Unlike prior work treating properties merely as static validation, PGS establishes them as feedback signals that actively navigate code correction, exploring broader utilization of properties in code repair.
> > >
> > >
> > > | **Related Work** | **Core Idea** | **Key Difference from PGS** |
> > > |------|-----------|-------------------------|
> > > | **QuickCheck**: A Lightweight Tool for Random Testing of Haskell Programs | Classic Property-Based Testing that finds counterexamples via manually defined properties and random sampling. | QuickCheck requires human experts to predefine formal properties; PGS achieves autonomous reasoning and extraction of properties from natural language requirements. Additionally, QuickCheck's shrinking is value/type-based for human readability, while PGS employs token-based structural minimization specifically optimized for LLM attention mechanisms and context efficiency. |
> > > | **nl2postcond**: Can Large Language Models Transform Natural Language Intent into Formal Method Postconditions? | Uses LLMs to transform manually-written natural language descriptions into formal method postconditions, evaluating translation quality via test correctness and mutation testing. | Nl2postcond treats properties as "translation targets", measuring how accurately LLMs translate manually written natural language into formal assertions. Its performance is upper-bounded by the quality of reference natural language/code; furthermore, it does not address how assertions can be operationalized to guide code repair. In contrast, PGS shows properties actively guide code correction in closed-loop repair without ground truth. |
> > > | **APT**: LLM-based Unit Test Generation via Property Retrieval | Property retrieval-augmented unit test generation, analyzing "Given-When-Then" patterns and relationships with existing test cases in the codebase to generate new tests. | APT relies on existing high quality test cases in external codebase as reference, retrieving related tests from sibling classes, parent classes, and interface-implementing classes via RAG to generate new tests. In contrast, PGS demonstrates the capability to generate properties in the absence of ground truth, with a focus on using these properties to detect and repair errors in code. |
> > > | **ClassInvGen**: Class Invariant Synthesis using Large Language Models | LLM-driven invariant synthesis that co-generates class invariants and tests, filtering incorrect candidates based on current code execution results. Its goal is to generate specifications for code whose logic is already determined. | ClassInvGen evaluates and filters invariants based on the execution results of the original code, treating the current code's behavior as the "gold standard." PGS assumes the code is buggy and extracts "logical truth" from problem statements to correct the code. This inversion of trust is precisely what distinguishes repair from specification mining. |
> > >
> > >
> > > **Table E:** Comparison with Mentioned Related Work
> > >
> > >
> > > ---
> > >
> > >
> > > ### **Q8: Sufficiency of Property Validation  in Complex Settings**
> > >
> > > We believe our responses to **Q3** and **Q6** demonstrate that PGS-generated properties are **harmless** in complex settings, as incorrect properties rarely mislead the repair process due to our filtering mechanisms and iterative repair. The critical question is not safety, but **utility**: how actively these properties guide repair in complex settings.
> > >
> > > Statistically, **Main Results (Table 3)** and **Section 5.3.4 (Figure 6)** confirm that our properties effectively drive repair by converting latent bugs into actionable signals across diverse scenarios, including complex settings. Specifically, unlike traditional PBT confined to static assertions for bug detection, PGS guides the model to implement **active verification strategies** in certain cases when generating properties: for instance, triggering exhaustive methods like brute-force simulation on minimal inputs, trading computational cost for deeper semantic validation (such as **Figure 9** in appendix). These innovative forms go beyond simple assertions used in prior work, representing an exploration specific to LLM code repair rather than mere bug detection, enhancing property utility for code repair in complex problems.

---

### Official Review · Reviewer_LrAA · 2026-03-13

**Soundness:** 3
**Presentation:** 3
**Significance:** 3
**Originality:** 3
**Overall Recommendation:** 4
**Confidence:** 3

**Summary:**

This paper introduces Property-Generated Solver (PGS), a novel framework for LLM-based code refinement. PGS moves beyond traditional input–output (I/O) test feedback and instead provides high-quality, property-driven feedback. The framework is built on two key principles: feedback should be property-oriented, offering semantic guidance beyond simple I/O mismatches, and structurally minimal, reducing cognitive load by isolating the root causes of errors.

**Compliance With Llm Reviewing Policy:**

Affirmed.

**Key Questions For Authors:**

Please check my questions in the comments.

**Limitations:**

Yes

**Strengths And Weaknesses:**

Strengths:
1. I particularly appreciate how the authors shift the focus from the limitations of traditional I/O pair–based testing to more realistic testing scenarios. By introducing assert-based program constructs, the approach constructs unit tests with contextual information, which better reflect real-world debugging and verification processes.
2. PGS consistently outperforms strong baselines across diverse benchmarks, achieving 1.4×–1.6× higher fix rates. These results demonstrate both the effectiveness of the proposed framework and its strong generalizability across different tasks and datasets.

Weaknesses:
1. The paper introduces the problem of the scarcity of high-quality test cases, particularly the cycle-of-self-deception issue. However, it assumes the availability of test cases and does not include a test generation module. As a result, the proposed method does not directly address this problem in automated test generation settings.
2. The writing clarity could be improved. Several parts of the paper are difficult to follow (e.g., Section 3.2), and the presentation would benefit from clearer explanations and better organization.


Specifically, the detailed comments are summarized as follows:

1. This work does not include a module for test generation. As a result, the “cycle-of-self-deception” issue that often arises in automated test generation settings is not directly addressed. The paper would benefit from clarifying whether PGS is intended to mitigate this issue or operates under a different assumption where test cases are already available.
2. Section 3.2 is difficult to follow before the PGS framework is fully introduced. Some of the design choices in this section appear more like engineering parameter tuning rather than hypothesis-driven validation. Moreover, the conclusions drawn in this section seem to hold primarily within the PGS framework itself, yet they are presented with broader claims of generality.
3. The baselines evaluated across datasets appear inconsistent. For example: Code-T does not report results on MBPP and MGDebugger is evaluated only on MBPP. Additionally, in the Ablation section, the authors present results of Self-Edit and MGDebugger on HumanEval, while these methods do not appear with HumanEval results in Table 3. This inconsistency makes it difficult to interpret the comparisons.
4. The ablation study on Performance vs. Token Consumption includes only a small number of baselines. The authors should clarify why Self-Edit and MGDebugger were selected. It would strengthen the analysis if additional methods were included. Furthermore, evaluating the approach on other models (e.g., DeepSeek-V3.1) could help demonstrate the robustness and general applicability of the method.
5. The paper does not sufficiently position the proposed feedback granularity design within existing literature. In particular, prior work that refines high-level I/O feedback using LLMs is not discussed or compared (e.g., Olausson et al., ICLR 2024). Moreover, it remains unclear how PGS compares with such LLM-refined feedback approaches.

[1]. Olausson, Theo X., et al. “Is Self-Repair a Silver Bullet for Code Generation?” ICLR 2024.

Other minor suggestions:
1. The abbreviation LLMs should be defined when it first appears in the paper (i.e., Large Language Models).
2. In Figure 3 caption, there is a repeated “the”.

---

> ### Author Rebuttal · Authors · 2026-03-31
>
> ### **Q1: PGS's Solution to "Cycle-of-Self-Deception" and Scarcity of Test Cases**
>
> - **How does PGS Address the "cycle-of-self-deception" Issue?** To clarify, PGS does not rely on oracle prediction (notoriously difficult and error-prone) to solve "cycle-of-self-deception", nor is it intended to leverage oracle prediction to address this issue. We argue that there exists other solutions beyond oracle prediction.
>
>   - Alternatively, we propose property validation to compensate for "cycle-of-self-deception". Property validation requires **property generation (lines 279-294)** and **input synthesis (lines 295-308)**, both of which are core components of PGS. Instead of predicting expected oracle, PGS checks for property violations, thereby eliminating the oracle prediction module entirely.
>
>   - The results demonstrate that property-guided feedback effectively mitigates "cycle-of-self-deception" issues while avoiding the fragility of oracle prediction.
>
> - **Assumption of Availability of Test Cases.** To clarify, PGS does not assume the availability of test cases. Consistent with previous methods, PGS only uses public test cases for property filtering. If there are no public test cases, PGS can also generate inputs through input synthesis to perform property checks (PGS does not require oracle prediction).
>
> ---
>
> ### **Q2: Writing Clarity of Section 3.2**
>
> We agree that Section 3.2 could be improved. In the revised manuscript, we will refine the organization to enhance clarity. Specifically, we will retain the validation of property validity in Section 3.1, while moving the delta-debugging ablation analysis to the Ablation Study section to ensure a more focused and logical flow.
>
> ---
>
> ### **Q3: Inconsistency in Reported Baseline Results**
>
> Due to space constraints, **Table 3** initially included only a subset of baselines, while complete results (including MGDebugger on HumanEval and Code-T on MBPP) were provided in **Appendix Table 5 (Lines 936-988)**.
>
> To eliminate confusion, we will move all baselines that appeared in the ablation study from **Appendix Table 5** into **Table 3**. We will also add explicit guiding text in the main body and **Table 3** caption and ensure all citations in the ablation study align with the updated tables.
>
> ---
>
> ### **Q4: Baseline Choices on Cost Analysis Ablation**
>
> - **Why Self-Edit and MGDebugger?** Self-Edit and MGDebugger were selected as they represent the examples of "simple resampling" and "complex multi-step debugging" respectively, sufficient to demonstrate PGS's trade-off between accuracy and token efficiency.
>
> - **Cost Analysis on More Baselines and Datasets.** We have conducted cost analysis using another model (i.e., DeepSeek-R1-Distilled-32B) in **Appendix D.5 (Lines 874-913, Figure 7)**. Moreover, we also provide results on more benchmarks, e.g., LiveCodeBench, MBPP and CodeContest (See **Table A** in **Reviewer D4no Q1** for details).
>
> We will add these results to the revised manuscript.
>
> ---
>
> ### **Q5: Related Works**
>
> We will cite and discuss these relevant studies in the revised manuscript.
>
> - Regarding Olausson et al. (2024) [1], their study suggests that resampling often outperforms refining due to the low quality of feedback in the refinement loop. This aligns with our observation in the "Introduction", i.e., previous TDD methods rely on scarce high-quality test cases, resulting in a lack of precise and actionable feedback during refinement.
>
> - PGS addresses this bottleneck by introducting a property-oriented and structurally minimal feedback mechanism. Through shifting from noisy execution signals to property-based verification, PGS significantly enhances the actionability of feedback, thereby unlocking the potential of the refinement paradigm.
>
> [1]. Olausson, Theo X., et al. “Is Self-Repair a Silver Bullet for Code Generation?” ICLR 2024.
>
> ---
>
> ### **Q6: Minor writing suggestions**
>
> Thanks for your advice. We will modify the revised version.

---

### Official Review · Reviewer_D4no · 2026-03-13

**Soundness:** 3
**Presentation:** 3
**Significance:** 3
**Originality:** 3
**Overall Recommendation:** 4
**Confidence:** 3

**Summary:**

This paper proposes PGS, a new method to help LLMs fix their own code bugs. Instead of using many test cases, PGS gives the model short, clear feedback based on simple rules (properties) and the easiest failing example.

**Compliance With Llm Reviewing Policy:**

Affirmed.

**Key Questions For Authors:**

See weaknesses

**Limitations:**

yes

**Strengths And Weaknesses:**

### Strengths
- Comprehensive experiments: The authors test PGS on many benchmarks, from simple function-level tasks (HumanEval, MBPP) to hard competition problems (LiveCodeBench) and real-world repo-level fixes (SWE-Bench), showing the method works well in different settings.
- Clear design principles: The paper introduces two easy-to-understand ideas—"property-oriented" feedback (checking high-level rules) and "structurally minimal" feedback (using the simplest failing input)—and shows through pilot studies why these help LLMs debug better.
- Strong empirical results: PGS achieves new state-of-the-art results across multiple models and datasets, with a bug fix rate 1.4x–1.6x higher than the best existing debugging methods.

### Weaknesses
- Token cost fairness: In Table 3, PGS uses more tokens per iteration than baselines due to property generation, but the table does not show "Average Token Cost per Problem" for each method. Adding this column would help readers judge if the performance gain is worth the extra cost. Also, Figure 7 only shows cost analysis for LiveCodeBench; adding more datasets (e.g., MBPP, CodeContest) would strengthen the cost-effectiveness claim.
- Ablation study: While the paper ablates core components (minimization, property generation, filtering), it does not fully isolate the contribution of "input synthesis" (generating new test inputs) in the main ablation table. A direct comparison of "PGS without input synthesis" vs. full PGS in Table 7 would make the analysis more complete.

---

> ### Author Rebuttal · Authors · 2026-03-31
>
> ### **Q1: Token cost fairness**
>
> We provide the cost-effectiveness analysis across multiple datasets in **Table A** below.
>
> - **Superior Token Effectiveness.** PGS achieves superior effectiveness on token cost compared with its counterparts. For example, on CodeContest, PGS at Iteration 3 achieves a higher accuracy (39.6%) than both Self-Edit (34.4%) and Self-Debugger (36.1%) at Iteration 5 , while consuming similar or significantly fewer tokens (PGS 31.9k vs. Self-Edit 32.0k vs. Self-Debugger 63.4k).
>
> - **Results on MBPP and CodeContest.** We provide additional cost analysis on MBPP and CodeContest in **Table A**. These results consistently follow the same trend as observed in LiveCodeBench.
>
> We will add these analyses (including the "Average Token Cost per Problem" column and MBPP/CodeContest results) to the revised manuscript.
>
> | Dataset | Method | Iteration 1(Acc/Cost Token) | Iteration 3 (Acc/Cost Token) | Iteration 5 (Acc/Cost Token) |
> | :--- | :--- | :---: | :---: | :---: |
> | **LiveCodeBench** | Self-Edit | 56.0% / 6.2k | 58.5% / 11.3k | 60.2% / 16.1k |
> | | Self-Debugger | 57.9% / 11.6k | 60.4% / 25.5k | 61.5% / 39.8k |
> | | **PGS (Ours)** | **59.3% / 9.6k** | **62.9% / 17.4k** | **65.1% / 24.2k** |
> | **MBPP** | Self-Edit | 75.7% / 3.5k | 77.1% / 7.6k | 78.2% / 11.8k |
> | | Self-Debugger | 76.2% / 9.8k | 78.9% / 19.1k | 80.0% / 30.7k |
> | | **PGS (Ours)** | **77.8% / 6.2k** | **81.1% / 12.3k** | **82.5% / 17.5k** |
> | **CodeContest** | Self-Edit | 32.5% / 10.7k | 33.8% / 21.5k | 34.4% / 32.0k |
> | | Self-Debugger | 33.2% / 19.4k | 35.7% / 42.6k | 36.1% / 63.4k |
> | | **PGS (Ours)** | **35.1% / 16.9k** | **39.6% / 31.9k** | **41.7% / 46.8k** |
>
> Table A: Cost-effectiveness analysis in **Qwen3-30B** across multiple benchmarks
>
> ---
>
> ### **Q2: Contribution of "Input Synthesis" in Component Ablation**
>
> We add "Input Synthesis" into our component ablation study (**Table 7** in paper). The results are present in **Table B** below.
>
> - **Ablation Setup.** In the "+Property Generation" setting, we do not synthesize additional inputs but only provide property-based feedback derived from existing public test cases. In the "+Input Synthesis" setting, we additionally synthesize extra inputs and provide property-based feedback grounded on the execution of these new inputs.
>
> - **Impact of "Input Synthese".** As shown in **Table B** below, Pass@1 improves from 68.5% to 69.4% (+0.9%), confirming that concrete counter-examples help the model identify logic errors. While "input synthesis" is not our core contribution (we adopt existing input synthesis techniques [1]), but our ablation validates its utility.
>
> We will add these results to the revised manuscript.
>
> [1] OpenAI. "Competitive Programming with Large Reasoning Models."  arXiv preprint arXiv:2502.06807
>
>
> | Method / Component | Pass@1 (%) | ∆ Improvement |
> | :--- | :---: | :---: |
> | Baseline (Raw I/O Feedback) | 64.4 | 0 |
> | + Structural Minimization | 67.2 | +2.8 |
> | + Property Generation | 68.5 | +1.3 |
> | + Input Synthesis | 69.4 | +0.9 |
> | + Property Check (Filtering) | 71.6 | +2.2 |
> | **+ Iterative Refinement (Full PGS)** | **76.5** | **+4.9** |
>
> Table B: Contribution of Each Component in PGS

---

> > ### Author Rebuttal · Reviewer_D4no · 2026-04-07
> >
> > The authors addressed most of ny concerns. After reading other reviews, I decided to keep my original score.

---

> > > ### Author Response · Authors · 2026-04-07
> > >
> > > We appreciate your constructive feedback and are glad that our rebuttal addressed your concerns. As suggested, we will add (1) the "Average Token Cost per Problem" column to the main results table, (2) cost-effectiveness curves for MBPP and CodeContest, and (3) the input synthesis ablation to the component analysis. These additions will strengthen the evaluation as you noted. Thank you for supporting our work.

---

### Decision · Program_Chairs · 2026-04-30

**Decision:**

Accept (regular)

**Comment:**

This paper was received positively overall. Reviewers agreed that the central intuition is compelling: in LLM code refinement, the quality of the feedback matters at least as much as the number of tests. The paper was also viewed as empirically strong, with broad evaluation across benchmarks and models, a useful pilot study, and consistent improvements over competitive refinement baselines.

The main remaining concerns were about how strongly to view the contribution as conceptually novel, and about the sufficiency of property validation in more complex settings. I think these are fair caveats, but the rebuttal addressed many of the practical questions well by clarifying the role of input synthesis, adding further cost-effectiveness and ablation evidence, and providing additional analysis of property accuracy, failure modes, and repository-level behavior. On balance, the paper meets the ICML bar, and I recommend acceptance.

For the final version, the authors should incorporate the additional analyses promised in the rebuttal, position the work more carefully relative to prior software engineering and property-based testing literature, and strengthen the discussion of limitations around property correctness, filtering, and cost.